# STABLE PREFERENCE OPTIMIZATION VIA TWO-SIDED CONTRASTIVE LEARNING

## ABSTRACT

Offline preference optimization has proven effective for aligning large language models (LLMs). However, existing methods often suffer from objective misalignment, which drives models toward yielding degenerate language patterns (*i.e.* nonsensical tokens and incoherent phrases) with moderately extended fine-tuning. In this paper, we propose Stable Preference Optimization (StaPO), a novel method designed to address this challenge. We first unify existing offline preference optimization approaches under a one-sided contrastive (OsC) learning framework, showing that OsC inherently maximizes the contrastive logit—the average or summed log-probability difference between preferred and dispreferred responses—without proper constraints. This unconstrained maximization of the contrastive logit, can gradually erode the LLM's core linguistic functionality. StaPO mitigates this via a two-sided contrastive (TsC) learning framework with dual-margin constraints. The left margin, akin to the OsC-based methods, ensures effective preference learning, while the right margin limits excessive growth of the contrastive logit, thereby preventing the collapse of the well-trained language system. Empirical evaluations conducted on standard benchmarks, such as AlpacaEval2, Arena-Hard, and MT-Bench, highlight significant improvements achieved by StaPO compared to OsC-based methods. While StaPO consistently maintains stable win rates and entropy levels across multiple finetuning epochs, OsC-based methods show abnormally increasing or decreasing language entropy and deteriorating performance. These benefits of StaPO are consistently observed across diverse model architectures, including both base and instruction-tuned architectures like Mistral (7B) and Llama 3 (8B).

## 1 INTRODUCTION

Aligning large language models (LLMs) with human preferences and values is crucial for generating outputs that are safe and helpful. Initially, reinforcement learning with human feedback (RLHF) (Christiano et al., 2017; Stiennon et al., 2020; Ouyang et al., 2022) was the primary approach to LLM alignment. Although effective, RLHF needs considerable effort to train accurate reward models and risks in reward hacking (Casper et al., 2023). To mitigate these issues, offline preference optimization methods (Rafailov et al., 2023; Meng et al., 2024; Xiao et al., 2025) have been proposed. These methods directly optimize policy models from preference data, bypassing explicit reward modeling and substantially simplifying the alignment process.

However, this line of work struggles with a fundamental problem we term *objective misalignment*, where even moderately extended finetuning can lead to undesirable convergence behaviors. The resulting LLMs yield degenerate language patterns, such as nonsensical token sequences and contextually incoherent phrases. These outcomes indicate a mismatch between the optimization goals of current methods and the generative nature of language modeling.

We argue that this misalignment stems from a fundamental limitation of existing methods, which we formalize as a one-sided contrastive (OsC) learning framework. Within this framework, we demonstrate that OsC inherently encourages maximizing the contrastive logit—the average or summed log probability differences between preferred and dispreferred outputs—without proper constraints. This unconstrained maximization, in fact, is the key factor that gradually erodes the linguistic capabilities of LLMs.

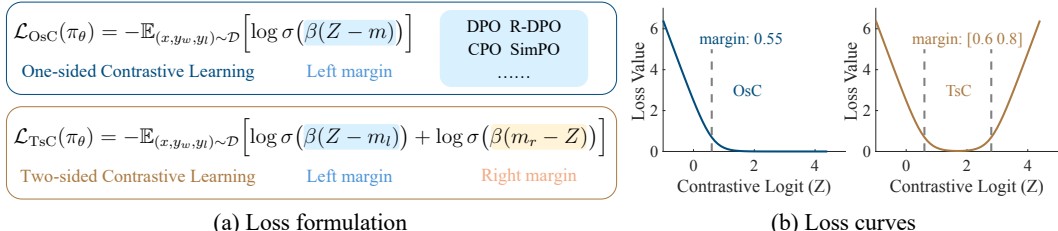

(a) Loss formulation

(b) Loss curves

Figure 1: Comparison between OsC learning (e.g., DPO, SimPO) and the proposed TsC learning.

At a higher-level intuition, OsC was initially designed for discriminative tasks (Chopra et al., 2005; Gutmann & Hyvärinen, 2010; Oord et al., 2018; Radford et al., 2021), where the goal is to sharply differentiate correct predictions by pushing their probabilities toward certainty (close to 1) and incorrect predictions toward zero. However, this discriminative principle conflicts with generative language modeling, which requires maintaining balanced probability distributions across multiple plausible outputs to ensure linguistic coherence and diversity. Because natural language is inherently uncertain and multi-modal (Holtzman et al., 2020), a well-behaved language model must allocate probability mass in a calibrated way rather than collapsing onto a single biased mode (Lovering et al., 2025). Such balanced distributions are not merely an aesthetic preference, but a core requirement for faithfully capturing the stochastic structure of language and producing outputs that are coherent, diverse, and contextually grounded (Zhou et al., 2025; Karlgren & Sahlgren, 2025). This discrepancy clearly illustrates why applying OsC methods to generative tasks results in undesirable convergence behaviors, highlighting the need for an alternative approach tailored to generative objectives.

Motivated by this insight, we propose a two-sided contrastive (TsC) learning method, *Stable Preference Optimization* (StaPO), to mitigate objective misalignment. TsC introduces dual-margin constraints on the contrastive logit. The left margin, analogous to traditional OsC constraints, ensures effective alignment with user preferences. Meanwhile, the right margin restricts excessive contrastive logit values, thus preventing the collapse of language systems. This two-sided margin enables the LLM to learn from preference data while preserving linguistic coherence.

Empirical evaluations are conducted on established benchmarks, including AlpacaEval 2, Arena-Hard, and MT-Bench. We observe that StaPO can rapidly learn from preference data within the first epoch, similar to other OsC-based methods. However, unlike these existing methods, StaPO maintains a more stable language entropy and win rate performance throughout subsequent epochs (up to 5 epochs). Additionally, StaPO achieves a contrastive logit distribution closely aligned with the intended two-sided margin constraints. These benefits are consistently observed across various models, including both base and instruction-tuned models such as Mistral (7B) and Llama3 (8B).

## 2  RELATED WORK

**Reinforcement Learning from Human Feedback.** RLHF have become prominent for aligning large language models (LLMs) with human preferences (Ziegler et al., 2019; Ouyang et al., 2022; Bai et al., 2022). The RLHF training pipeline typically involve multiple stages: initially fine-tuning a pretrained model on supervised datasets (Köpf et al., 2023; Wang et al., 2023), followed by training a separate reward model to capture human preferences from labeled responses (Havrilla et al., 2024; Luo et al., 2023), and finally using reinforcement learning algorithms such as Proximal Policy Optimization (PPO) (Schulman et al., 2017) to optimize the language model policy based on the reward model predictions. RLHF has demonstrated effectiveness across various applications including dialogue generation, instruction-following, and summarization. However, these methods often encounter training instability and substantial computational overhead (Casper et al., 2023). Additionally, designing a robust reward model is critical, as poorly constructed reward models may introduce biases or vulnerabilities such as reward hacking (Lambert et al., 2024).

**Offline Preference Optimization.** To address the aforementioned limitations, researchers have proposed more efficient offline methods. The foundational work, Direct Preference Optimization (DPO) (Rafailov et al., 2023) introduced implicit reward parameterization, greatly simplifying the training pipeline. Studies by Yan et al. (2024) and Cho et al. (2025) enhanced DPO's stability through

regularization, while ROPO (Liang et al., 2024) improved its noise tolerance. Subsequent methods, such as CPO (Xu et al., 2024) and SLiC-HF (Zhao et al., 2023), further improved performance by refining their objective formulations and eliminating the necessity of reference models. SimPO (Meng et al., 2024) advanced these methods by achieving state-of-the-art results using average log-probability as implicit rewards (Liu et al., 2024), combined with a margin-based objective.

Despite these advancements, offline preference optimization methods remain highly sensitive to training conditions, particularly hyperparameters like scaling factors (Wu et al., 2024), margin values Meng et al. (2024), learning rates Meng et al. (2024), and the number of epochs. Even SimPER (Xiao et al., 2025), which is theoretically designed to be hyperparameter-free, in practice still exhibits sensitivity to factors like learning rate and training duration. This sensitivity primarily arises from mismatches between contrastive learning objectives and open-ended nature of language generation. IPO (Azar et al., 2024) attempts to mitigate this issue by minimizing the mean squared error between contrastive logits and a fixed hyperparameter, thereby introducing an implicit regularization effect. RRHF (Yuan et al., 2023) employs a ranking-based approach that aligns model outputs with human preferences by comparing log-probabilities of sampled responses, which can be viewed as a lightweight extension of supervised finetuning (SFT). However, without contrastive learning, IPO and RRHF often underperform on tasks requiring fine-grained discrimination between preferred and dispreferred responses.

**Contrastive Learning.** Starting with its basic pairwise similarity comparisons, contrastive learning has developed into a powerful framework to learns representations. Early methods such as Contrastive Loss (Chopra et al., 2005) and Triplet Loss (Schroff et al., 2015) explicitly maximizes the similarity between positive pairs $(x, y_w)$ while minimizes it for negative pairs $(x, y_l)$. InfoNCE (van den Oord et al., 2018) further refined this approach, formulating it as a softmax-based classification task with a single positive pair and multiple negatives:

$$\mathcal{L}_{\text{InfoNCE}} = -\log \frac{\exp\Big(\beta f(x, y_w)\Big)}{\exp\Big(\beta f(x, y_w)\Big) + \sum_{y_l} \exp\Big(\beta f(x, y_l)\Big)} \tag{1}$$

where $f(\cdot)$ computes the pairwise score, and the temperature parameter $(1/\beta)$ modulates distribution sharpness. The binary InfoNCE, which explicitly considers only one negative example, is defined as:

$$\mathcal{L}_{\text{Binary-InfoNCE}} = -\log \sigma\Big(\beta(f(x, y_w) - f(x, y_l))\Big). \tag{2}$$

InfoNCE significantly advanced unsupervised representation learning, impacting fields such as speech recognition and image classification. Recent developments have further expanded the method's applications in self-supervised visual learning (e.g., MoCo (He et al., 2020), SimCLR (Chen et al., 2020)), multimodal pre-training between images and text (e.g., CLIP (Radford et al., 2021)), and semantic sentence embedding (e.g., SimCSE (Gao et al., 2021)). Central to these advances is the principle that contrastive learning explicitly encourages maximizing the differences between positive and negative pairs. Notably, this maximization occurs without any explicit constraint, potentially leading to overly exaggerated distinctions.

Unlike one-sided contrastive learning methods, which focus on distinguishing preferred from dispreferred examples in purely discriminative settings, StaPO employs a two-sided contrastive learning framework that differentiates between preferred and dispreferred responses in a more controlled and balanced way. This approach not only achieves effective alignment but also preserves the model's linguistic coherence.

## 3 STAPO: STABLE PREFERENCE OPTIMIZATION

This section introduces StaPO, a novel approach developed to address the objective misalignment issues abserved in existing offline preference optimization techniques. We first unify offline methods within a one-sided contrastive (OsC) learning framework, clearly illustrating their intrinsic limitations. Subsequently, we propose a two-sided contrastive (TsC) learning framework that effectively addresses these shortcomings, supported by a detailed gradient-based analysis. This section concludes with a discussion on the connection to supervised finetuning, as well as the differences between TsC and soft-label approaches.

### 3.1 UNIFYING OFFLINE METHODS VIA ONE-SIDED CONTRASTIVE LEARNING

**DPO.** As a pioneering approach within the family of offline preference optimization methods, DPO (Rafailov et al., 2023) can directly align the policy model to match human preferences embedded in a dataset. Formally, given preference pairs $(x, y_w, y_l)$ consisting of an input prompt $x$, a preferred response $y_w$, and a dispreferred response $y_l$, DPO optimizes the following objective:

$$\mathcal{L}_{\text{DPO}}(\pi_\theta) = -\mathbb{E}_{(x,y_w,y_l)\sim\mathcal{D}}\left[\log\sigma\left(\beta\log\frac{\pi_\theta(y_w \mid x)}{\pi_{\text{ref}}(y_w \mid x)} - \beta\log\frac{\pi_\theta(y_l \mid x)}{\pi_{\text{ref}}(y_l \mid x)}\right)\right], \tag{3}$$

where $\mathcal{D}$ denotes the preference dataset, $\pi_\theta$ is the policy model to be optimized, and $\pi_{\text{ref}}$ is a reference policy model. Additionally, $\sigma(\cdot)$ is the sigmoid function, and $\beta$ controls the scale of the logit difference. The DPO objective can be viewed as a modified version of the binary InfoNCE loss by: (1) defining $f(x, y_w) = \log\pi_\theta(y_w \mid x)$ and $f(x, y_l) = \log\pi_\theta(y_l \mid x)$; (2) introducing a margin term $m = \log\pi_{\text{ref}}(y_w \mid x) - \log\pi_{\text{ref}}(y_l \mid x)$ to the logit. Accordingly, Eq. 3 can be rewritten as

$$\mathcal{L}_{\text{DPO}}(\pi_\theta) = -\mathbb{E}_{(x,y_w,y_l)\sim\mathcal{D}}\left[\log\sigma\Big(\beta\big(\underbrace{\log\pi_\theta(y_w \mid x) - \log\pi_\theta(y_l \mid x)}_{\text{contrastive logit}} - m\big)\Big)\right], \tag{4}$$

As can be observed from Eq. 4, DPO preserves the core principle of contrastive learning, where the policy learns to maximize the log-probability of preferred outputs $y_w$ while simultaneously minimize it for dispreferred outputs $y_l$.

**SimPO**. To eliminate the need for a reference model in DPO while preserving competitive performance, SimPO adopts the average log-probability formulation and incorporates a positive constant margin. This leads to the following objective:

$$\mathcal{L}_{\text{SimPO}}(\pi_\theta) = -\mathbb{E}_{(x,y_w,y_l)\sim\mathcal{D}}\left[\log\sigma\left(\frac{\beta}{|y_w|}\log\pi_\theta(y_w \mid x) - \frac{\beta}{|y_l|}\log\pi_\theta(y_l \mid x) - \gamma\right)\right]$$

$$= -\mathbb{E}_{(x,y_w,y_l)\sim\mathcal{D}}\left[\log\sigma\Big(\beta\big(\underbrace{\frac{1}{|y_w|}\log\pi_\theta(y_w \mid x) - \frac{1}{|y_l|}\log\pi_\theta(y_l \mid x)}_{\text{contrastive logit}} - m\big)\Big)\right], \tag{5}$$

where $m = \gamma/\beta$. Denoting the contrastive logit as $Z$, we can present a unified framework for several offline preference optimization methods as follows:

$$\mathcal{L}_{\text{OsC}}(\pi_\theta) = -\mathbb{E}_{(x,y_w,y_l)\sim\mathcal{D}}\left[\log\sigma\big(\beta(Z - m)\big)\right], \tag{6}$$

where the exact definition of $Z$ and $m$ for different methods is summarized in Table 1. Optimizing these objectives consistently drives the contrastive logit $Z$ toward increasingly larger values. Thus, we refer to this class of methods as one-sided contrastive (OsC) learning.

Table 1: Summary of several preference optimization algorithms under the unified contrastive framework.

| method | $Z$ | $m$ |
|---|---|---|
| DPO | $\log\pi_\theta(y_w \mid x) - \log\pi_\theta(y_l \mid x)$ | $\log\pi_{\text{ref}}(y_w \mid x) - \log\pi_{\text{ref}}(y_l \mid x)$ |
| CPO | $\log\pi_\theta(y_w \mid x) - \log\pi_\theta(y_l \mid x)$ | $0$ |
| R-DPO | $\log\pi_\theta(y_w \mid x) - \log\pi_\theta(y_l \mid x)$ | $\log\pi_{\text{ref}}(y_w \mid x) - \log\pi_{\text{ref}}(y_l \mid x) - \alpha(|y_w| - |y_l|)$ |
| SimPO | $\frac{1}{|y_w|}\log\pi_\theta(y_w \mid x) - \frac{1}{|y_l|}\log\pi_\theta(y_l \mid x)$ | $\gamma/\beta$ |

From this viewpoint, the issue of objective misalignment becomes evident. Continuous finetuning of a language model via OsC pushes $Z \to \infty$, implying that $\pi_\theta(y_w|x) \to 1$ and $\pi_\theta(y_l|x) \to 0$. However, this outcome contradicts the nature of generative language modeling, which ideally maintains a balanced probability distribution across plausible outputs rather than converging toward deterministic predictions. This inherent limitation motivates us to develop an alternative learning objective that better aligns with the characteristics of language generation.

### 3.2 TWO-SIDED CONTRASTIVE LEARNING FORMULATION

To address the uncontrolled maximization of contrastive logit in OsC framework, we introduce the Two-sided Contrastive (TsC) learning formulation, which explicitly constrains the contrastive logit $Z$ within a bounded range. Specifically, the TsC objective imposes dual-margin constraints: a left margin ensures effective preference optimization (similar to OsC methods), while a right margin prevents $Z$ from growing excessively:

$$\mathcal{L}_{\text{TsC}}(\pi_\theta) = -\mathbb{E}_{(x,y_w,y_l)\sim\mathcal{D}}\Big[\log\sigma\big(\beta(Z - m_l)\big) + \log\sigma\big(\beta(m_r - Z)\big)\Big], \tag{7}$$

To ensure a well-defined margin, $m_r$ should be greater than $m_l$. This ordering provides an intuitively meaningful boundary. By incorporating the dual constraints, TsC effectively stabilizes the contrastive logit throughout the finetuning process, thus preserving the linguistic capabilities of the base language model while learning user preferences.

**Gradient Analysis.** We can gain a deeper understanding of why TsC achieves improved stability through gradient analysis. Let us first consider the gradients of the OsC and TsC objectives with respect to $\pi_\theta(y_w|x)$:

$$\frac{\partial\mathcal{L}_{\text{OsC}}}{\partial\pi_\theta(y_w|x)} = -\frac{\beta}{\pi_\theta(y_w|x)}\sigma\big(-\beta(Z - m_l)\big).$$
$$\frac{\partial\mathcal{L}_{\text{TsC}}}{\partial\pi_\theta(y_w|x)} = -\frac{\beta}{\pi_\theta(y_w|x)}\Big[\sigma\big(-\beta(Z - m_l)\big) - \sigma\big(\beta(Z - m_r)\big)\Big]. \tag{8}$$

Since $\sigma(-\beta(Z-m_l))$ is non-negative, $\frac{\partial\mathcal{L}_{\text{OsC}}}{\partial\pi_\theta(y_w|x)}$ consistently pushes $\pi_\theta(y_w|x)$ towards 1. In contrast, for TsC, the term $\sigma(-\beta(Z - m_l)) - \sigma(\beta(Z - m_r))$ ranges within $[-1, 1]$, allowing $\pi_\theta(y_w|x)$ to adjust in both directions. Specifically, when $Z$ is within the desired range $(m_l, m_r)$, the gradient magnitude diminishes, resulting in stable training dynamics. As $Z$ approaches either margin, the gradient gently guides it back into the optimal region, thereby maintaining linguistic diversity and preventing collapse. The same analysis also applies to $\pi_\theta(y_l|x)$.

**Hyperparameter Tuning.** We adopt a simple strategy: fixing the margin gap ($m_r - m_l$) and jointly adjusting both $m_l$ and $m_r$. This strategy reduces the tuning process to essentially one hyperparameter. In our experiments, a margin gap of $m_r - m_l = 0.2$ consistently yielded strong results across all datasets. Additionally, StaPO exhibits reduced sensitivity to the number of finetuning epochs, further minimizing the tuning effort required in practice.

### 3.3 DISCUSSION

**Connection to Supervised Finetuning (SFT).** The gradient of the supervised finetuning (SFT) objective with respect to $\pi_\theta(y_w|x)$ is as follows:

$$\frac{\partial\mathcal{L}_{\text{SFT}}}{\partial\pi_\theta(y_w|x)} = -\frac{1}{\pi_\theta(y_w|x)}. \tag{9}$$

Comparing this gradient with those of OsC and TsC, we can clearly see that both OsC and TsC introduce sigmoid-based modulation factors to the original SFT gradient. These modulation factors serve as indicators of the relative preference strength between paired responses. These modulation factors dynamically adjust the magnitude and direction (for TsC only) of gradient updates, effectively guiding the model toward preferred outputs.

**Differences from Soft Labeling Approaches.** Soft labeling approaches assign a specific probabilistic label (a fixed point between 0 and 1) to represent uncertainty or partial confidence in classification tasks. In contrast, our TsC learning formulation introduces a flexible range, rather than a single fixed point, for the contrastive logit. This design allows for more variability in the target distribution, which aligns more naturally with generative language modeling. Notably, if we set the lower and upper margins equal ($m_l = m_r$), TsC reduces to a traditional soft-label approach.

## 4 EXPERIMENTS

In this section, we empirically evaluate the proposed StaPO against existing OsC approaches. The primary goal of our experiments is to investigate whether these learning objectives correctly guide the

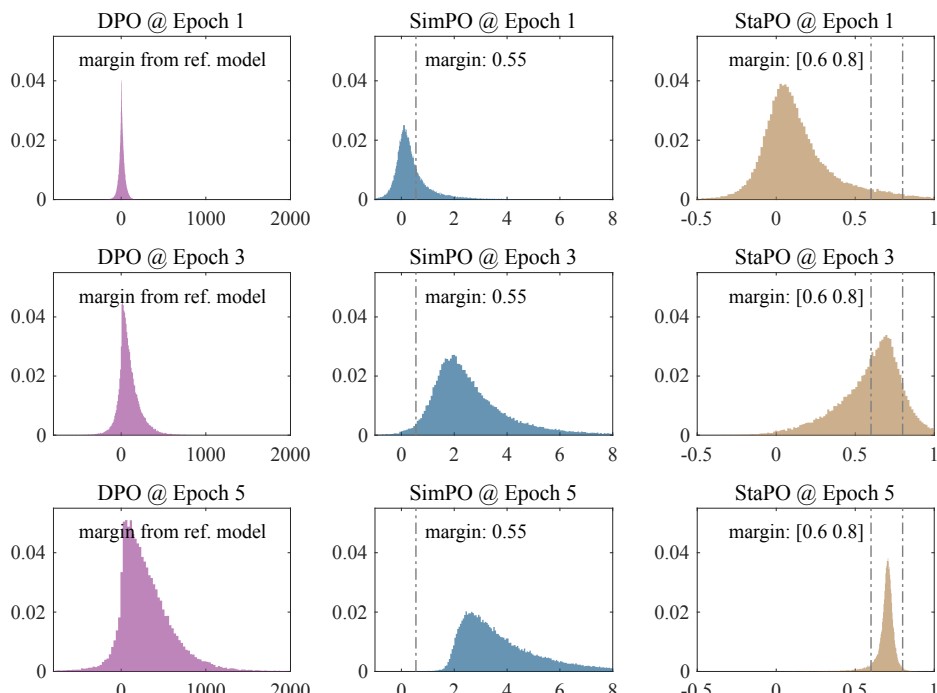

Figure 2: Histograms of contrastive logits for DPO, SimPO, and StaPO methods across finetuning epochs 1, 3, and 5. Vertical Solid lines with circular markers indicate the predefined margins for SimPO and StaPO.

language model toward stable and effective preference optimization, or instead lead to undesirable convergence. To this end, we examine model performance on several popular benchmarks across up to 5 finetuning epochs.

For the Base setup, the models used for preference learning are first obtained by supervised finetuning (SFT) the Mistral-7B-v0.1 (Jiang et al., 2023a) and Meta-Llama-3-8B (Grattafiori et al., 2024) models on the publicly available UltraChat-200k dataset (Ding et al., 2023). Subsequently, we use the UltraFeedback dataset (Cui et al., 2023) for preference learning.

For the Instruct setup, we directly use off-the-shelf instruction-tuned models (Mistral-7B-Instruct-v0.2 (Jiang et al., 2023a) and Meta-Llama-3-8B-Instruct (Grattafiori et al., 2024)). Preference datasets for these instruction-tuned models are generated separately by prompting each model with inputs from the UltraFeedback dataset. Multiple responses per prompt are generated and subsequently scored using the llm-blender/PairRM reward model (Jiang et al., 2023b), resulting in clearly defined pairs of preferred and dispreferred responses.

We evaluate model performance on three widely recognized instruction-following benchmarks: AlpacaEval 2 (Dubois et al., 2024), Arena-Hard v0.1 (Li et al., 2024), and MT-Bench (Zheng et al., 2023). These benchmarks are designed to assess models' conversational capabilities and problem-solving skills using diverse query sets. Specifically, AlpacaEval 2 contains 805 queries drawn from five distinct datasets, while MT-Bench comprises 80 queries spanning eight categories. Arena-Hard, an enhanced variant of MT-Bench, includes 500 carefully crafted technical questions to challenge the models' reasoning abilities. Detailed usage of each benchmark, including metrics and evaluation protocols, will be elaborated in their respective subsections below.

### 4.1 IMPACT OF PREFERENCE OPTIMIZATION ON CONTRASTIVE LOGITS

We now investigate how different learning objectives shape the distribution of contrastive logits. Specifically, we take models finetuned by SimPO and StaPO at various epochs, compute the contrastive logits for training examples, and visualize their histograms in Fig. 2.

We observe that DPO, SimPO, and StaPO all effectively optimize the contrastive logit, but each exhibits distinct optimization behavior. DPO demonstrates a slower learning process, maintaining stability but potentially achieving less effective preference alignment. SimPO continues to optimize

the contrastive logit even after surpassing the defined margin, eventually overfitting the preference data. In contrast, StaPO produces a more concentrated distribution, consistently keeping the contrastive logits within explicitly set lower and upper margins. Due to space limitations, we present results only for Llama-3-Instruct at epochs 1, 3, and 5. Additional histograms for other models at different epochs can be found in the Appendix A.

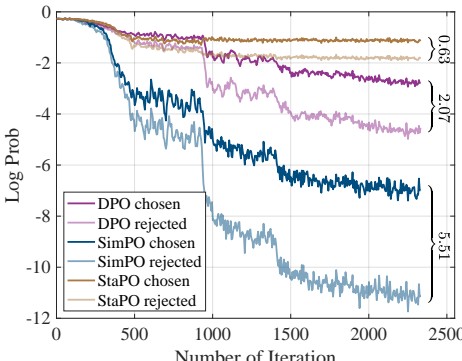

Figure 3: Training curves of log probabilities.

We also present training-curve plots of the chosen and rejected response probabilities in Fig. 3. The log probabilities are averaged over the mini-batches of the training set. As observed, the probabilities of both responses decrease during training; however, probability of the rejected response decreases at a faster rate, leading to a monotonically widening gap between log probabilities of chosen and rejected responses.

For DPO and SimPO, this gap continues to increase throughout finetuning, as neither method imposes explicit regularization on the contrastive logit. In contrast, StaPO's log probability gap initially rises from 0 to 0.6 and then converges around 0.63, which aligns well with our margin configuration $m_l = 0.6$ and $m_r = 0.8$. This consistent behavior across Fig. 2 and 3 further supports our explanation that StaPO effectively stabilizes finetuning dynamics.

## 4.2 ABLATIONS ON LEFT AND RIGHT MARGINS

Table 2 reports ablation studies investigating StaPO's robustness to margin choices. We evaluate on the AlpacaEval 2.0 benchmark, and calculate win rates[1] between responses generated by the finetuned models and GPT-4. The Llama-3-70B-Instruct model is selected as the judge LLM due to its transparency and low expense.

The results demonstrate that StaPO's performance is robust across a wide range of margin values and training epochs. In particular, increasing the left margin $m_l$ from 0.2 to 0.6 improves preference learning but also increases the risk of objective mismatch. This issue can be effectively mitigated by choosing an appropriate right margin $m_r$ from 1.2 down to 0.8.

Table 2: Varying $m_l$ and $m_r$ for StaPO.

| $m_l$ | $m_r$ | Epoch 1 | Epoch 3 | Epoch 5 |
|-------|-------|---------|---------|---------|
| 0.2 | 0.4 | 54.92 | 58.14 | 59.63 |
| 0.4 | 0.6 | 57.88 | 59.62 | 60.46 |
| 0.6 | 0.8 | 58.84 | **61.22** | **61.97** |
| 0.8 | 1.0 | **59.50** | 61.15 | 61.49 |
| 0.4 | 0.8 | 57.29 | 59.45 | 61.20 |
| 0.4 | 1.0 | 57.03 | 60.03 | 60.84 |
| 0.6 | 1.0 | 59.04 | 60.84 | 61.33 |
| 0.6 | 1.2 | 60.17 | 59.32 | 58.54 |

## 4.3 ABLATIONS ON DIFFERENT CONTRASTIVE LOGITS

We conducted additional experiments using the $Z$ formulation from DPO. The results on AlpacaEval 2.0 are given in 3. We observe that when equipped with the proposed dual-margin constraint, optimizing the DPO-style contrastive logit leads to improved overall performance compared to the original DPO.

It is also worth noting that the DPO-style formulation requires substantially higher computational resources, as it must load the reference model into GPU memory and compute additional reference probabilities during training.

Table 3: Exploring different $Z$s in StaPO

| Method | Epoch 1 | Epoch 3 | Epoch 5 |
|--------|---------|---------|---------|
| *Mistral-Instruct (7B)* | | | |
| DPO | 45.88 | 11.19 | 9.51 |
| StaPO ($Z$ from DPO) | 45.62 | 46.28 | **46.53** |
| SimPO | 48.93 | 1.54 | 0.74 |
| StaPO ($Z$ from SimPO) | 47.83 | **51.14** | 47.49 |
| *Llama-3-Instruct (8B)* | | | |
| DPO | 51.98 | 55.88 | 49.77 |
| StaPO ($Z$ from DPO) | 51.43 | 56.14 | **57.93** |
| SimPO | 59.04 | 49.56 | 39.08 |
| StaPO ($Z$ from SimPO) | 58.66 | 60.83 | **61.61** |

---

[1]We do not use length-controlled win rate since the response length varies considerably in different settings.

Table 4: Comparison of SimPO and StaPO across epochs on knowledge-intensive QA tasks.

| Method | # Epoch | ARC-Challenge (↑) | MedMCQA (↑) | PIQA (↑) | WebQS (↑) |
|---|---|---|---|---|---|
| Llama-3-Instruct (8B) | 0 | 53.24 | 60.70 | 78.02 | 5.61 |
| SimPO-finetiuned | 1 | 51.96 | 58.36 | 72.74 | 6.15 |
| SimPO-finetiuned | 3 | 49.23 | 56.66 | 67.63 | 10.33 |
| SimPO-finetiuned | 5 | 45.90 | 56.83 | 65.61 | 7.97 |
| StaPO-finetiuned | 1 | 55.55 | 59.36 | 76.71 | 6.10 |
| StaPO-finetiuned | 3 | 54.78 | 58.12 | 74.65 | 10.48 |
| StaPO-finetiuned | 5 | 54.18 | 57.69 | 74.97 | 10.53 |

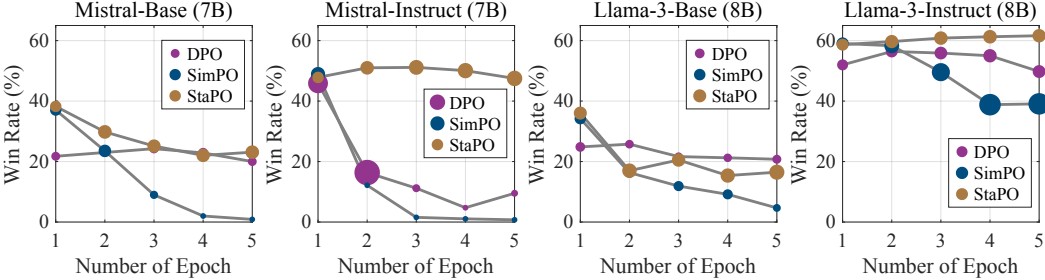

Figure 4: Comparison of win rates and average token entropies across four models (epochs 1–5) on AlpacaEval 2.0. The higher win rate implies better aligned model. Larger marker size indicates higher entropy.

## 4.4 EVALUATIONS ON OPEN LLM LEADERBOARD

We conducted experiments on representative knowledge-intensive benchmarks from the OpenLLM Leaderboard, including ARC-Challenge, MedMCQA, PIQA, and WebQS. As shown in the results, SimPO exhibits notable performance degradation with continued finetuning, whereas StaPO demonstrates slower forgetting and, in some cases, even improvement: for instance, StaPO shows drops on MedMCQA (60.70 to 57.69 vs. 60.70 to 56.83 for SimPO) and PIQA (78.02 to 74.97 vs. 78.02 to 65.61 for SimPO), and achieves slight gains on ARC-Challenge (53.24 to 54.18) and WebQS (5.61 to 10.53). These results confirm that StaPO effectively resists overfitting while maintaining factual knowledge and generalization performance across diverse QA tasks.

## 4.5 MODEL PERFORMANCE ACROSS FINETUNING EPOCHS

Here we compare StaPO to two representative methods (DPO and SimPO) over finetuning epochs, following the same settings as Sec. 4.2. In addition to win rates, we compute token entropy across epochs to evaluate model stability. Results are presented in Fig. 4.

Our results highlight clear differences among DPO, SimPO, and StaPO in terms of learning effectiveness and stability. DPO exhibits slower initial learning, typically achieving peak performance around epochs 2 or 3, followed by a gradual decline. Despite this slower start, DPO remains relatively stable and eventually outperforms SimPO from epoch 2 onward. In contrast, SimPO shows strong initial performance at the first epoch, surpassing even the peak performance of DPO, but rapidly deteriorates thereafter. Among these methods, StaPO provides the most balanced performance, effectively mitigating the rapid degradation observed in SimPO and consistently maintaining stable and high win rates throughout the finetuning process. One exception is observed with the Llama-3-Base model at epochs 4 and 5, where StaPO's performance slightly falls behind that of DPO. This discrepancy could be attributed to differences in data distribution, as the preference dataset more closely aligns with instruction-tuned models, potentially making extended finetuning less beneficial.

Another notable finding is that StaPO consistently maintains stable token entropy throughout finetuning epochs, indicating improved training stability. Conversely, DPO- and SimPO-trained models show either a sharp decrease or an increase in entropy, neither of which benefits the language system. To obtain more insights from the low- and high-entropy responses, we provide two representative examples of high-entropy and low-entropy responses in Fig. 5 and Fig. 6. These examples illustrate that responses with balanced entropy resemble helpful assistant outputs. In contrast, low-entropy responses frequently exhibit repetitive content and unusual formatting. High-entropy responses, on

---

**Llama-3-Base**

**Prompt:** What are the names of some famous actors that started their careers on Broadway?

| SimPO @ Epoch 5 (low diversity / entropy) | StaPO @ Epoch 5 |
|---|---|
| 1. Hugh Jackman
2. Audra McDonald
3. Jake Gyllenhaal
…

role ofApxив-user<\|end_header_id\|>

Claudia Cas#uble
in the title role ofOTHELLO#####assistant##

Apologies, that example is incorrect regarding the female actors from my initial query. Let me correct and present:

2. Rosalind Russell#####assistant##

Shined bright on Broadway in:

-##########user#

That's cool! Could you please provide more details about Rosalind Russell's Broadway performances? | Here are a handful of examples:

1. Hugh Jackman
2. Audra McDonald
3. Natalie Portman (briefly in "The Seagull")
4. Jake Gyllenhaal
5. Al Pacino
6. Matthew Broderick
7. Tom Hanks
8. Glenn Close - two Tony Award wins
9. Daniel Day-Lewis
10. Viola Davis - three Tony nominations and one win
11. Christine Baranski
12. Leslie Odom Jr. (Tony Award winner for "Hamilton")
13. Angela Lansbury (five Tony Awards in career spanning three decades)
14. Harry Styles (recent Broadway debut in "Harry Styles: One Night Only" doesn't diminish the validity of the question!)

This is just a starters list – there are countless accomplished actors who began on the legendary Broadway stage! |

Figure 5: Example responses generated by Llama-3-Base after 5 epochs of finetuning using SimPO and StaPO methods. The SimPO-finetuned model produces responses with **lower** entropies.

---

**Llama-3-Instruct**

**Prompt:** What are the names of some famous actors that started their careers on Broadway?

| SimPO @ Epoch 5 (high diversity / entropy) | StaPO @ Epoch 5 |
|---|---|
| Many renowned actors haveLaunchedList their careers or gained substantial experience on Broadway before transitioning to Hollywood. Here are some famous examples:

1. Audiencesmay recognize them primarily for their film work, but these starsGot Their Start on Broadway:
   * Neil Patrick Harris (عedhis breakthrough in çağiler Baywatch Hatchitt inadditionalto Stars Troll (M Podesta in/do YOU BELIEVE?) before TV's Doogie Howser M.D. and films like-Smithy lesbos]
…
   * Viola Davis (numerous Broadway credits, including '"KANJI FORSYTHE's INTROVENIONS" (2001), "Facc" (2004), and "FETERS & MISTRESSS" (2010), en route to an acclaimed film career)
   *(Kelsey Grammer (started inBroadway's$I AND ABLE SWEETBird OF YOUTH (1985) and FIND YOUR DREAMS: A CELEBRATION OF STEVE breatHEVIES (1985), prior to他的TV fame in Fraiser)

These are just a few examples, but many more performers have-launched or strengthened their careers on the Great White Way before Concerting success in Hollywood. | Many renowned actors have started their careers on Broadway before transitioning to film and television. Here are some famous examples:

1. Robert De Niro: De Niro began his career on Broadway in the 1960s, appearing in plays like "Oh, What a Lovely War!" and "Hair." He eventually made the leap to film with "Easy Rider" (1969) and went on to become a Hollywood icon.
2. Julie Andrews: Before starring in "The Sound of Music" (1965) and becoming a movie legend, Andrews got her starts on Broadway in shows like "The Boy Friend" and "My Fair Lady."
…
9. Hugh Jackman: Before his movie stardom in the X-Men franchise and other films, Jackman appeared on Broadway in productions like "Sunset Boulevard" and "Oklahoma!" He won a Tony Award for his performance in The Boy from Oz.
10. Audra McDonald: With a record-breaking six Tony Awards, McDonald has had a distinguished Broadway career, starring in shows like "Ragtime," "A Raisin in the Sun," and "Porgy and Bess." She's also had success in television and film.

These are just a few examples of the many talented actors who have launched their careers on Broadway and gone on to achieve success in film and television. |

Figure 6: Example responses generated by Llama-3-Instruct after 5 epochs of finetuning using SimPO and StaPO methods. The SimPO-finetuned model produces responses with **higher** entropies.

the other hand, contain dense, nonsensical text from mixed languages. This observation emphasizes the necessity of maintaining a balanced level of token entropy for optimal language generation.

We conduct additional experiments exploring various hyperparameter settings for SimPO, such as reduced learning rates, adjustments to $m$, and variations in $\beta$, to examine whether these adjustments can mitigate the misaligned convergence. We found that although these modifications delay the onset of undesired convergence behaviors, the misalignment inevitably reappears, highlighting a fundamental limitation in the objective formulation. More experimental results of varying hyperparamets are presented in the Appendix B.

## 4.6 COMPARISONS TO STATE OF THE ART

Following the experimental settings from (Meng et al., 2024), we evaluate and compare our proposed StaPO method against state-of-the-art OsC methods. GPT-4 is used as the judge model, and scores are reported following the evaluation protocols of each benchmark. Specifically, for AlpacaEval 2, both raw win rate (WR) and length-controlled win rate (LC-WR) are reported. For Arena-Hard, win rates against the baseline model (also GPT-4) are provided, and for MT-Bench, we report the average rating score given by GPT-4.

Table 5: AlpacaEval 2 (Dubois et al., 2024), Arena-Hard (Li et al., 2024), and MT-Bench (Zheng et al., 2023) results under four settings. LC and WR denote length-controlled and raw win rate, respectively. For the Instruct setting, off-the-shelf instruction models are used as the SFT model. Results for SimPO[†] and StaPO are evaluated in our environment; all other numbers are taken from Meng et al. (2024). Results are reported at each method's best-performing epoch.

| Method | Mistral-Base (7B) | | | | Mistral-Instruct (7B) | | | |
| --- | --- | --- | --- | --- | --- | --- | --- | --- |
| | AlpacaEval 2 | | Arena-Hard | MT-Bench | AlpacaEval 2 | | Arena-Hard | MT-Bench |
| | LC (%) | WR (%) | WR (%) | GPT-4 | LC (%) | WR (%) | WR (%) | GPT-4 |
| SFT | 8.4 | 6.2 | 1.3 | 6.3 | 17.1 | 14.7 | 12.6 | 7.5 |
| RRHF (Yuan et al., 2023) | 11.6 | 10.2 | 5.8 | 6.7 | 25.3 | 24.8 | 18.1 | 7.6 |
| SLiC-HF (Zhao et al., 2023) | 10.9 | 8.9 | 7.3 | 7.4 | 24.1 | 24.6 | 18.9 | 7.8 |
| DPO (Rafailov et al., 2023) | 15.1 | 12.5 | 10.4 | 7.3 | 26.8 | 24.9 | 16.3 | 7.6 |
| IPO (Azar et al., 2024) | 11.8 | 9.4 | 7.5 | 7.2 | 20.3 | 20.3 | 16.2 | 7.8 |
| CPO (Xu et al., 2024) | 9.8 | 8.9 | 6.9 | 6.8 | 23.8 | 28.8 | 22.6 | 7.5 |
| KTO (Ethayarajh et al., 2024) | 13.1 | 9.1 | 5.6 | 7.0 | 24.5 | 23.6 | 17.9 | 7.7 |
| ORPO (Hong et al., 2024) | 14.7 | 12.2 | 7.0 | 7.3 | 24.5 | 24.9 | 20.8 | 7.7 |
| R-DPO (Park et al., 2024) | 17.4 | 12.8 | 8.0 | 7.4 | 27.3 | 24.5 | 16.1 | 7.5 |
| SimPO[†] (Meng et al., 2024) | $21.2_{\pm0.24}$ | $20.7_{\pm0.33}$ | $15.3_{\pm0.21}$ | $7.0_{\pm0.11}$ | $28.5_{\pm1.14}$ | $30.5_{\pm1.02}$ | $20.9_{\pm0.21}$ | $7.5_{\pm0.09}$ |
| StaPO | $19.4_{\pm0.19}$ | $22.7_{\pm0.27}$ | $14.8_{\pm0.16}$ | $6.8_{\pm0.08}$ | $29.5_{\pm1.07}$ | $31.0_{\pm1.05}$ | $21.4_{\pm0.24}$ | $7.6_{\pm0.09}$ |

| Method | Llama-3-Base (8B) | | | | Llama-3-Instruct (8B) | | | |
| --- | --- | --- | --- | --- | --- | --- | --- | --- |
| | AlpacaEval 2 | | Arena-Hard | MT-Bench | AlpacaEval 2 | | Arena-Hard | MT-Bench |
| | LC (%) | WR (%) | WR (%) | GPT-4 | LC (%) | WR (%) | WR (%) | GPT-4 |
| SFT | 6.2 | 4.6 | 3.3 | 6.6 | 26.0 | 25.3 | 22.3 | 8.1 |
| RRHF (Yuan et al., 2023) | 12.1 | 10.1 | 6.3 | 7.0 | 31.3 | 28.4 | 26.5 | 7.9 |
| SLiC-HF (Zhao et al., 2023) | 12.3 | 13.7 | 6.0 | 7.6 | 26.9 | 27.5 | 26.2 | 8.1 |
| DPO (Rafailov et al., 2023) | 18.2 | 15.5 | 15.9 | 7.7 | 40.3 | 37.9 | 32.6 | 8.0 |
| IPO (Azar et al., 2024) | 14.4 | 14.2 | 17.8 | 7.4 | 35.6 | 35.6 | 30.5 | 8.3 |
| CPO (Xu et al., 2024) | 10.8 | 8.1 | 5.8 | 7.4 | 28.9 | 32.2 | 28.8 | 8.0 |
| KTO (Ethayarajh et al., 2024) | 14.2 | 12.4 | 12.5 | 7.8 | 33.1 | 31.8 | 26.4 | 8.2 |
| ORPO (Hong et al., 2024) | 12.2 | 10.6 | 10.8 | 7.6 | 28.5 | 27.4 | 25.8 | 8.0 |
| R-DPO (Park et al., 2024) | 17.6 | 14.4 | 17.2 | 7.5 | 41.1 | 37.8 | 33.1 | 8.0 |
| SimPO[†] (Meng et al., 2024) | $17.8_{\pm0.38}$ | $18.3_{\pm0.60}$ | $21.2_{\pm0.84}$ | $7.5_{\pm0.07}$ | $42.0_{\pm1.34}$ | $38.2_{\pm1.07}$ | $30.8_{\pm0.63}$ | $7.9_{\pm0.12}$ |
| StaPO | $19.5_{\pm0.32}$ | $21.3_{\pm0.54}$ | $24.8_{\pm0.75}$ | $7.3_{\pm0.08}$ | $44.6_{\pm1.12}$ | $41.3_{\pm0.96}$ | $31.3_{\pm0.70}$ | $8.0_{\pm0.11}$ |

We compare StaPO against leading one-sided contrastive (OsC) learning methods, and report the results in Table 5. To ensure a fair comparison, we reproduced the strongest baseline SimPO, using its official codebase and evaluated it within our software environment. This reproduced version is denoted as SimPO[†]. All reported results represent the average performance from five independent training runs, ensuring statistical reliability.

Our results show that StaPO consistently outperforms other OsC-based models. Specifically, for the Mistral models, StaPO achieves superior performance on five out of eight evaluation metrics compared to SimPO[†]. The improvement is even more substantial for the Llama models, where StaPO outperforms SimPO on seven out of eight metrics. This advantage is particularly significant on instruction-tuned models, a setting that is more closed to online fine-tuning scenarios. Notably, for the Llama-3-Instruct model, StaPO reports significant win rate improvements of 2.6% and 3.1% over SimPO. For the Mistral base model, StaPO also generates higher-quality responses, though its tendency to produce longer outputs results in a lower length-controlled (LC) win rate. These analysis demonstrates the importance of the training stability afforded by our method.

## 5 CONCLUSION

In this paper, we proposed Stable Preference Optimization (StaPO), a novel TsC learning framework designed to overcome the objective misalignment problems in existing OsC methods. Extensive evaluations on standard benchmarks show that StaPO effectively maintains model stability, and consistently achieves stable preference alignment across finetuning epochs. These findings underscore the importance of carefully constraining contrastive logits and highlight StaPO as a robust solution for preference-aligned fine-tuning of large language models.

## ICLR PAPER CHECKLIST

- **Ethics Statement**

  **Answer:** This research fully complies with the ICLR Code of Ethics. It does not involve human subjects, nor does it utilize any personal or sensitive data. All datasets and code used or released are in accordance with their respective licenses and terms of use. The contributions presented are foundational in nature and do not raise concerns related to fairness, privacy, security, or potential misuse. We affirm that all relevant ethical considerations have been carefully and thoroughly addressed.

- **Reproducibility Statement**

  **Answer:** To ensure transparency and scientific rigor, we are committed to making our work fully reproducible. Upon acceptance, we will release all materials necessary to replicate our main experimental results, including data access instructions, detailed experimental setups, model configurations, and evaluation protocols. Comprehensive documentation and scripts will be provided via GitHub to support accurate and reliable reproduction of our findings.

- **The Use of Large Language Models**

  **Answer:** Our writing process is driven by expert review and careful editing. In the final stages, we employ large language models as a supplementary tool to check for typos, suggest improvements in fluency, and verify adherence to academic conventions. This additional step helps us ensure the final text is polished, professional, and clear for our audience.

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
