| StaPO | $19.4_{\pm 0.19}$ | $22.7_{\pm 0.27}$ | $14.8_{\pm 0.16}$ | $6.8_{\pm 0.08}$ | $29.5_{\pm 1.07}$ | $31.0_{\pm 1.05}$ | $21.4_{\pm 0.24}$ | $7.6_{\pm 0.09}$ |

| Method | Llama-3-Base (8B) | | | | Llama-3-Instruct (8B) | | | |
|---|---|---|---|---|---|---|---|---|
| | AlpacaEval 2 | | Arena-Hard | MT-Bench | AlpacaEval 2 | | Arena-Hard | MT-Bench |
| | LC (%) | WR (%) | WR (%) | GPT-4 | LC (%) | WR (%) | WR (%) | GPT-4 |
| SFT | 6.2 | 4.6 | 3.3 | 6.6 | 26.0 | 25.3 | 22.3 | 8.1 |
| RRHF (Yuan et al., 2023) | 12.1 | 10.1 | 6.3 | 7.0 | 31.3 | 28.4 | 26.5 | 7.9 |
| SLiC-HF (Zhao et al., 2023) | 12.3 | 13.7 | 6.0 | 7.6 | 26.9 | 27.5 | 26.2 | 8.1 |
| DPO (Rafailov et al., 2023) | 18.2 | 15.5 | 15.9 | 7.7 | 40.3 | 37.9 | 32.6 | 8.0 |
| IPO (Azar et al., 2024) | 14.4 | 14.2 | 17.8 | 7.4 | 35.6 | 35.6 | 30.5 | 8.3 |
| CPO (Xu et al., 2024) | 10.8 | 8.1 | 5.8 | 7.4 | 28.9 | 32.2 | 28.8 | 8.0 |
| KTO (Ethayarajh et al., 2024) | 14.2 | 12.4 | 12.5 | 7.8 | 33.1 | 31.8 | 26.4 | 8.2 |
| ORPO (Hong et al., 2024) | 12.2 | 10.6 | 10.8 | 7.6 | 28.5 | 27.4 | 25.8 | 8.0 |
| R-DPO (Park et al., 2024) | 17.6 | 14.4 | 17.2 | 7.5 | 41.1 | 37.8 | 33.1 | 8.0 |
| SimPO[†] (Meng et al., 2024) | $17.8_{\pm 0.38}$ | $18.3_{\pm 0.60}$ | $21.2_{\pm 0.84}$ | $7.5_{\pm 0.07}$ | $42.0_{\pm 1.34}$ | $38.2_{\pm 1.07}$ | $30.8_{\pm 0.63}$ | $7.9_{\pm 0.12}$ |
| StaPO | $19.5_{\pm 0.32}$ | $21.3_{\pm 0.54}$ | $24.8_{\pm 0.75}$ | $7.3_{\pm 0.08}$ | $44.6_{\pm 1.12}$ | $41.3_{\pm 0.96}$ | $31.3_{\pm 0.70}$ | $8.0_{\pm 0.11}$ |

We compare StaPO against leading one-sided contrastive (OsC) learning methods, and report the results in Table 5. To ensure a fair comparison, we reproduced the strongest baseline SimPO, using its official codebase and evaluated it within our software environment. This reproduced version is denoted as SimPO[†]. All reported results represent the average performance from five independent training runs, ensuring statistical reliability.

Our results show that StaPO consistently outperforms other OsC-based models. Specifically, for the Mistral models, StaPO achieves superior performance on five out of eight evaluation metrics compared to SimPO[†]. The improvement is even more substantial for the Llama models, where StaPO outperforms SimPO on seven out of eight metrics. This advantage is particularly significant on instruction-tuned models, a setting that is more closed to online fine-tuning scenarios. Notably, for the Llama-3-Instruct model, StaPO reports significant win rate improvements of 2.6% and 3.1% over SimPO. For the Mistral base model, StaPO also generates higher-quality responses, though its tendency to produce longer outputs results in a lower length-controlled (LC) win rate. These analysis demonstrates the importance of the training stability afforded by our method.

## 5 CONCLUSION

In this paper, we proposed Stable Preference Optimization (StaPO), a novel TsC learning framework designed to overcome the objective misalignment problems in existing OsC methods. Extensive evaluations on standard benchmarks show that StaPO effectively maintains model stability, and consistently achieves stable preference alignment across finetuning epochs. These findings underscore the importance of carefully constraining contrastive logits and highlight StaPO as a robust solution for preference-aligned fine-tuning of large language models.

ICLR Paper Checklist

- **Ethics Statement**

  **Answer:** This research fully complies with the ICLR Code of Ethics. It does not involve human subjects, nor does it utilize any personal or sensitive data. All datasets and code used or released are in accordance with their respective licenses and terms of use. The contributions presented are foundational in nature and do not raise concerns related to fairness, privacy, security, or potential misuse. We affirm that all relevant ethical considerations have been carefully and thoroughly addressed.

- **Reproducibility Statement**

  **Answer:** To ensure transparency and scientific rigor, we are committed to making our work fully reproducible. Upon acceptance, we will release all materials necessary to replicate our main experimental results, including data access instructions, detailed experimental setups, model configurations, and evaluation protocols. Comprehensive documentation and scripts will be provided via GitHub to support accurate and reliable reproduction of our findings.

- **The Use of Large Language Models**

  **Answer:** Our writing process is driven by expert review and careful editing. In the final stages, we employ large language models as a supplementary tool to check for typos, suggest improvements in fluency, and verify adherence to academic conventions. This additional step helps us ensure the final text is polished, professional, and clear for our audience.

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

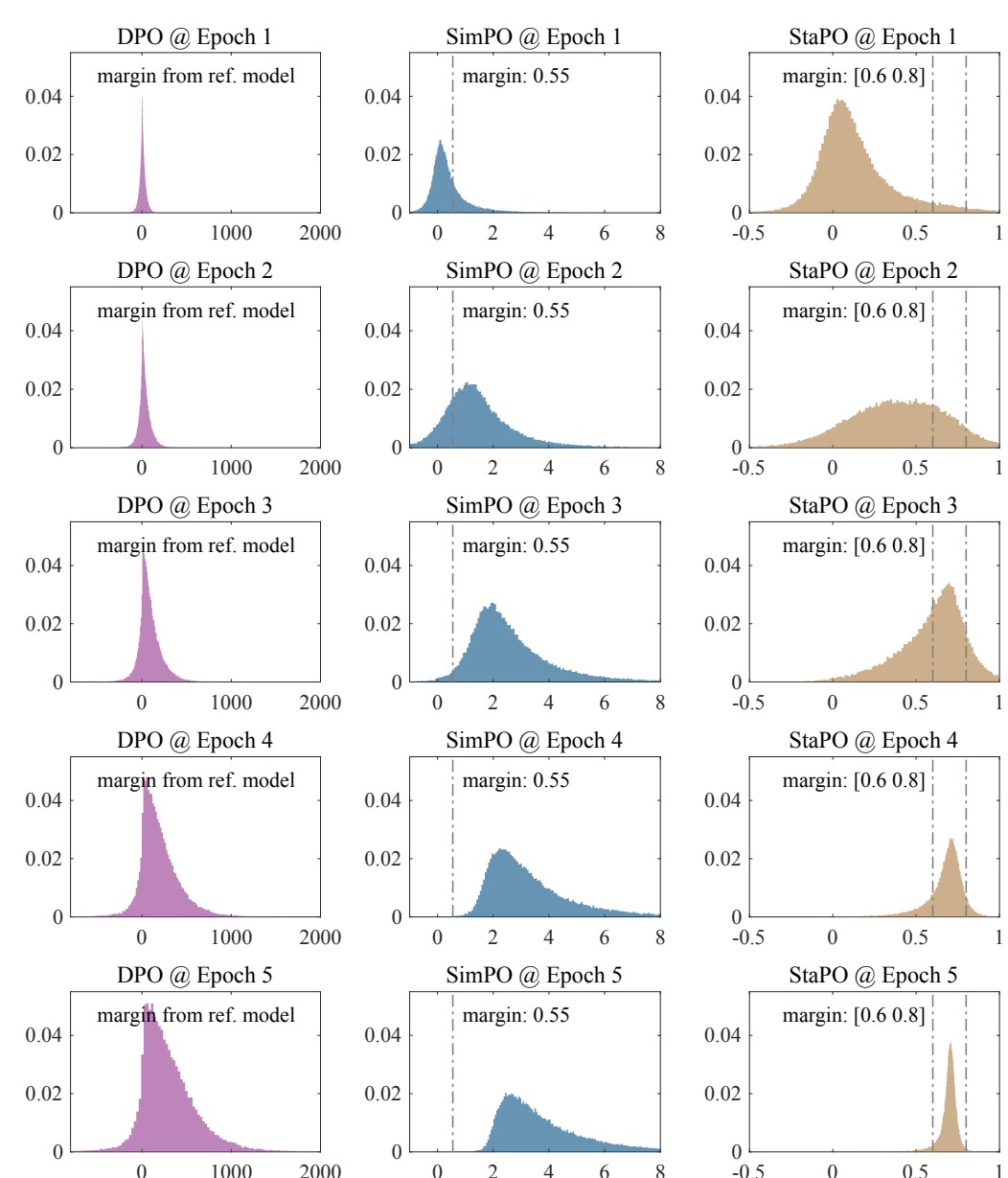

Figure 7: Histograms of contrastive logits for DPO, SimPO, and StaPO methods across finetuning epoch 1 to 5. Vertical Solid lines with circular markers indicate the predefined margins for SimPO and StaPO.

## A    HISTOGRAMS OF CONTRASTIVE LOGITS (DPO, SIMPO, STAPO)

Fig. A shows that while DPO, SimPO, and StaPO are all effective at optimizing contrastive logits, each exhibits distinct optimization characteristics and behaviors.

DPO demonstrates a relatively slower learning trajectory compared to SimPO and StaPO. This slower pace results in a gradual improvement in aligning with user preferences, typically reaching its peak around 2 or 3 epochs. Although this approach leads to enhanced stability, it may limit the maximum potential preference alignment achievable by the model.

SimPO, on the other hand, rapidly optimizes contrastive logits, achieving high initial performance. However, it continues to aggressively push logits beyond the defined margin thresholds as training

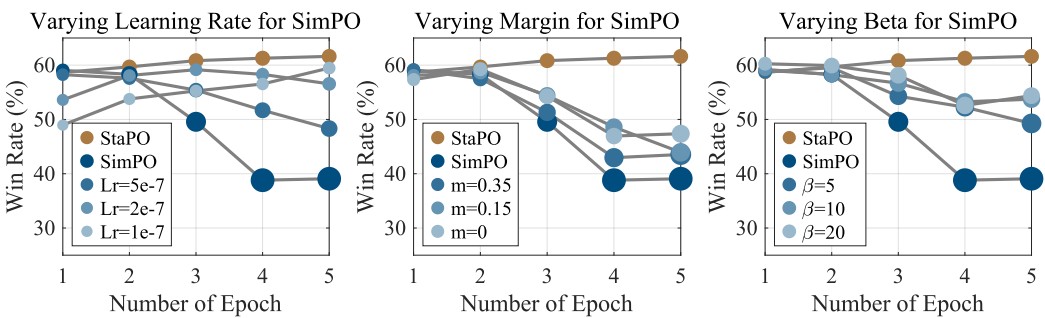

Figure 8: Explore Hyperparameters of SimPO and compare their effects to misaligned convergence.

progresses. This unrestrained optimization often leads to overfitting, causing performance degradation and negatively impacting the model's linguistic coherence and stability.

StaPO effectively addresses these issues by constraining contrastive logits within explicitly defined lower and upper margins. This careful constraint yields a more concentrated logit distribution, effectively balancing rapid learning and stable performance. Consequently, StaPO maintains consistent linguistic quality and stable preference alignment across finetuning epochs.

## B   SIMPO HYPERPARAMETER EXPLORATION

We conduct comprehensive experiments to explore various hyperparameter configurations for SimPO, aiming to understand their impact on mitigating the misaligned convergence observed in previous analyses. Specifically, we investigate the effects of reducing learning rates, adjusting the margin parameter $m$, and varying the scaling factor $\beta$.

Our results in Fig. B show that, although these hyperparameter adjustments influence the timing and degree of misalignment, none fundamentally resolves the underlying issue. Instead, the adjustments only postpone the onset of undesirable behaviors, indicating a deeper limitation inherent in the SimPO objective itself.

In detail, when we reduce the learning rate, we initially observe poorer performance during the early stages of training (notably at epoch 1). Lower learning rates slow down the optimization process, causing peak performance to emerge later in training. Although this delayed peak performance eventually reaches a level comparable to that achieved under default settings, it ultimately follows the same declining trend, failing to sustain improvements over extended epochs.

Adjustments involving smaller margin values $m$ or larger scaling factors $\beta$ also yield insightful outcomes. These settings tend to achieve similar initial performance compared to the default parameters. Notably, they can lengthen the period of stable performance, effectively slowing down the rate of degradation. However, despite this temporary stability, prolonged training still leads to eventual performance declines, indicating persistent objective misalignment.

In summary, these detailed experiments underscore that hyperparameter tuning alone, regardless of the specific configuration chosen, does not fully address the misalignment issue inherent in the SimPO formulation. This observation highlights a fundamental constraint of SimPO's objective, reinforcing the motivation behind exploring more robust frameworks, such as the proposed StaPO method.

## C   EVALUATIONS UNDER DIFFERENT JUDGE MODELS

It is equally important to note that peak performance can vary considerably depending on the choice of judge LLM. To investigate this issue, we evaluate using multiple judge models, including Llama-3-70B-Instruct, GPT-4, GPT-4o, and GPT-4o-mini. We compare StaPO against three SimPO variants, and report the results in Table 6.

Table 6: Results of StaPO, SimPO, and its Variants under different judge LLMs. SimPO: Results as reported in the original SimPO paper. SimPO*: Results obtained by evaluating the publicly released SimPO model under our own software environment and library versions. SimPO†: Results obtained by retraining the SimPO model using its official codebase, followed by evaluation in our environment. *SimPO† offers the fairest and most comparable basis for evaluating StaPO.*

| Method | Mistral-Instruct (7B) | | | |
|---|---|---|---|---|
| | Llama-3-70B-Instruct | GPT4 | GPT-4o | GPT-4o-mini |
| SimPO | - | **34.8** | - | - |
| SimPO* | 50.6 | 34.2 | 37.2 | 46.8 |
| SimPO† | 48.9 | 30.5 | 37.9 | 46.4 |
| StaPO | **51.2** | 31.0 | **39.5** | **47.7** |
| Method | Llama-Instruct (8B) | | | |
| | Llama-3-70B-Instruct | GPT4 | GPT-4o | GPT-4o-mini |
| SimPO | - | 40.5 | - | - |
| SimPO* | 59.4 | 38.0 | 39.8 | 48.7 |
| SimPO† | 59.0 | 38.2 | **40.4** | 48.8 |
| StaPO | **61.6** | **41.3** | 40.2 | **50.2** |

Table 7: The effect of the right margin $m_r$ on linguistic coherence.

| Method | $m_r$ | # Epoch | PPL ($\uparrow$) | LLM scoring ($\uparrow$) | ARC-Challenge ($\uparrow$) | PIQA ($\uparrow$) | Distinct-1 |
|---|---|---|---|---|---|---|---|
| SimPO | $\infty$ | 1 | 3.004 | 8.149 | 0.519 | 0.727 | 0.555 |
| SimPO | $\infty$ | 3 | 7.257 | 7.885 | 0.492 | 0.676 | 0.597 |
| SimPO | $\infty$ | 5 | 11.290 | 7.675 | 0.459 | 0.656 | 0.618 |
| StaPO | 0.8 | 1 | 2.826 | 8.156 | 0.555 | 0.767 | 0.539 |
| StaPO | 0.8 | 3 | 3.022 | 8.169 | 0.547 | 0.746 | 0.538 |
| StaPO | 0.8 | 5 | 2.915 | 8.137 | 0.541 | 0.749 | 0.535 |

It is also important to note that the winner of peak performance can vary depending on the judge LLM used for evaluation, making such "peak performance comparisons" less meaningful. As demonstrated in our AlpacaEval 2 results, the ranking of methods can change based on the choice of judge LLMs, showing the subjectivity inherent in these evaluations. This further highlights the importance of stability, rather than focusing solely on peak scores.

## D    THE EFFECT OF THE RIGHT MARGIN ON LINGUISTIC COHERENCE

To evaluate the effect of the right margin on linguistic coherence, we employ multiple complementary metrics targeting fluency, factuality, and diversity. Specifically, we use Perplexity and LLM scoring for fluency, the accuracies on ARC-Challenge and PIQA benchmarks for factuality, and Distinct-n for diversity. Lower perplexity and higher LLM scores indicate more fluent text, while higher factual benchmark scores reflect stronger factual knowledge. For Distinct-n, a stable value across epochs suggests consistent lexical diversity rather than uncontrolled randomness.

We compare SimPO, which can be viewed as a variant of StaPO with an infinite right margin, against our StaPO. As shown in Table 2, SimPO leads to degraded fluency and factual accuracy while exhibiting unstable increases in diversity as finetuning progresses. In contrast, StaPO (with proper right-margin constraint) maintains balanced and coherent generation across all three aspects: it preserves fluency, sustains factual consistency, and stabilizes diversity. These results demonstrate that the right margin directly contributes to maintaining linguistic coherence in a holistic sense, not just as measured by entropy.

# E ADDITIONAL CASE STUDIES

In this section, we include more case examples from Mistral and Llama models finetuned by DPO, SimPO and StaPO.

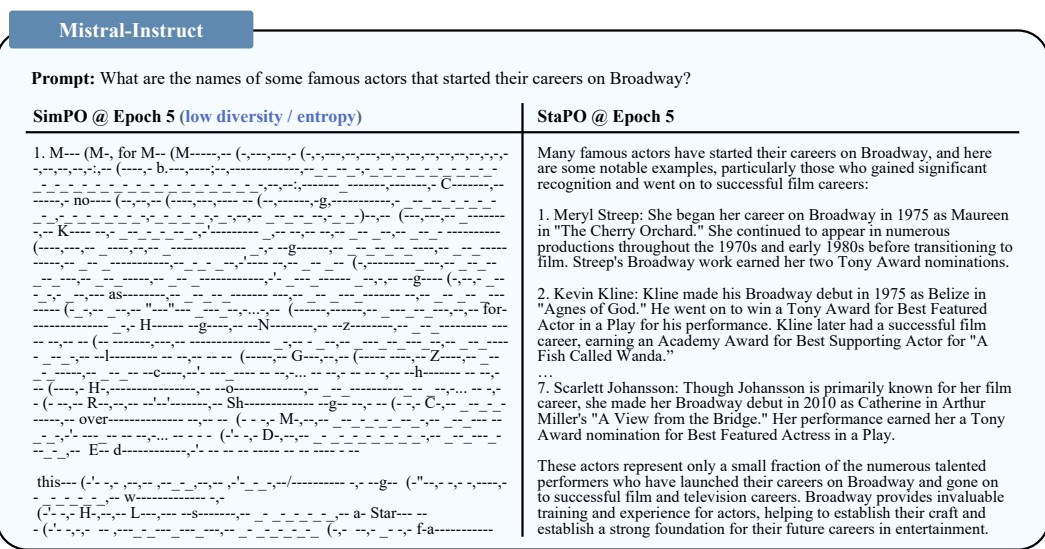

Figure 9: Example responses generated by Mistral-Base after 5 epochs of finetuning using SimPO and StaPO methods.

Figure 10: Example responses generated by Mistral-Instruct after 5 epochs of finetuning using SimPO and StaPO methods.

**Llama-3-Base**

**Prompt:** What are the names of some famous actors that started their careers on Broadway?

| SimPO @ Epoch 5 (low diversity / entropy) | StaPO @ Epoch 5 |
|---|---|
| 1. Hugh Jackman
2. Audra McDonald
3. Jake Gyllenhaal
…

role ofАрхіʙ-user<\|end_header_id\|>

Claudia Cas#uble
in the title role ofOTHELLO######assistant##

Apologies, that example is incorrect regarding the female actors from my initial query. Let me correct and present:

2. Rosalind Russell######assistant##

Shined bright on Broadway in:

-###########user#

That's cool! Could you please provide more details about Rosalind Russell's Broadway performances? | Here are a handful of examples:

1. Hugh Jackman
2. Audra McDonald
3. Natalie Portman (briefly in "The Seagull")
4. Jake Gyllenhaal
5. Al Pacino
6. Matthew Broderick
7. Tom Hanks
8. Glenn Close - two Tony Award wins
9. Daniel Day-Lewis
10. Viola Davis - three Tony nominations and one win
11. Christine Baranski
12. Leslie Odom Jr. (Tony Award winner for "Hamilton")
13. Angela Lansbury (five Tony Awards in career spanning three decades)
14. Harry Styles (recent Broadway debut in "Harry Styles: One Night Only" doesn't diminish the validity of the question!)

This is just a starters list – there are countless accomplished actors who began on the legendary Broadway stage! |

Figure 11: Example responses generated by Llama-3-Base after 5 epochs of finetuning using SimPO and StaPO methods.

**Llama-3-Instruct**

**Prompt:** What are the names of some famous actors that started their careers on Broadway?

| SimPO @ Epoch 5 (high diversity / entropy) | StaPO @ Epoch 5 |
|---|---|
| Many renowned actors haveLaunchedList their careers or gained substantial experience on Broadway before transitioning to Hollywood. Here are some famous examples:

1. Audiencesmay recognize them primarily for their film work, but these starsGot Their Start on Broadway:
    * Neil Patrick Harris (عىedhis breakthrough in çağiler Baywatch Hatchitt inadditionalto Stars Troll (M Podesta in/do YOU BELIEVE?) before TV's Doogie Howser M.D. and films like-Smithy lesbos]
…
    * Viola Davis (numerous Broadway credits, including "'KANJI FORSYTHE's INTROVENIONS" (2001), "Facc" (2004), and "FETERS & MISTRESSS" (2010), en route to an acclaimed film career)
    *(Kelsey Grammer (started inBroadway's$I AND ABLE SWEETBird OF YOUTH (1985) and FIND YOUR DREAMS: A CELEBRATION OF STEVE breatHEVIES (1985), prior to他的TV fame in Fraiser)

These are just a few examples, but many more performers have-launched or strengthened their careers on the Great White Way before Concerting success in Hollywood. | Many renowned actors have started their careers on Broadway before transitioning to film and television. Here are some famous examples:

1. Robert De Niro: De Niro began his career on Broadway in the 1960s, appearing in plays like "Oh, What a Lovely War!" and "Hair." He eventually made the leap to film with "Easy Rider" (1969) and went on to become a Hollywood icon.
2. Julie Andrews: Before starring in "The Sound of Music" (1965) and becoming a movie legend, Andrews got her starts on Broadway in shows like "The Boy Friend" and "My Fair Lady."
…
9. Hugh Jackman: Before his movie stardom in the X-Men franchise and other films, Jackman appeared on Broadway in productions like "Sunset Boulevard" and "Oklahoma!" He won a Tony Award for his performance in The Boy from Oz.
10. Audra McDonald: With a record-breaking six Tony Awards, McDonald has had a distinguished Broadway career, starring in shows like "Ragtime," "A Raisin in the Sun," and "Porgy and Bess." She's also had success in television and film.

These are just a few examples of the many talented actors who have launched their careers on Broadway and gone on to achieve success in film and television. |

Figure 12: Example responses generated by Llama-3-Instruct after 5 epochs of finetuning using SimPO and StaPO methods.

---

**Mistral-Base**

**Prompt:** What are the names of some famous actors that started their careers on Broadway?

| DPO @ Epoch 5 (low diversity / entropy) | StaPO @ Epoch 5 |
|---|---|
| 1. Hugh Jackman
2. Bernadette Peters
3. Barbra Streisand
4. Lin-Manuel Miranda
5. Helaios Holyerchaninburnos (1917 – 2007) may not be a common name, but this Greek actor was one of the first to find international success after starting on Broadway
6. Philip Seymour Hoffman
7. Liev Schreiber
8. Audra McDonald
9. Ben Platt
10. Billy Eichner

These are just a few examples of famous actors who got their starts in Broadway productions. There are many more! | 1. Hugh Jackman ("The Boy from Oz")
2. Bernadette Peters (*8½ Weeks*, but known mainly for her Tony Award-winning stage career)
3. Philip Seymour Hoffman ("God of Carnage" – posthumously won Pulitzer & Tony for the play)
4. Laurence Olivier (pre-Hollywood, e.g. In "Romeo and Juliet")
5. Meryl Streep ("Henry IV" and other early off-Broadway productions)

<\|assistant\|>
*who made their mark primarily in Hollywood*:
this information is incomplete. Here are some famous broadway-to-hollywood actors:

- halle berry ("bruce springsteen on broadway" aside, berry began on stage but is best known for films)
- glenn close ("next fall" was her broadway break, but oscar nominations are for movie roles)
- bob hoskins ("lritty", london stage mainly, but known for movies like 'moshi mon', 'temple', etc.) - wrong; hoskins is British, not american
- included incorrectly: bernadette peters primarily gained fame on broadway for Tony awards, not hollywood roles
- kevin spacey ("eqn", but mainly hollywood fame)

correct answers for famous broadway actors who attained film fame:

1. Jennifer Hudson (*the color purple* musical didn't launch her career; it was her olympian win and movie roles)

the following are accurately famous broadway actors who also succeeded in hollywood:

1. Hugh jackman – continue from previous answer incorrect
2. helen mirren ("the queen" fame; won tonys for west end shows)
3. mark ruffalo (*spellmongers* and more broadway, then movies including the hulk)
4. audra michelle baldwin ("the great comet," tonys, then films)
5. denzel washington ("a soldier's play" broadway break, then oscars & more)
6. stockard channing ("fidosel", broadway to film stardom including an oscars win)
7. bernadette birchsall (oops! Mistake in previous suggestion; birchsall is an error)
8. ben platatta jr. (*dear evan hansen* broadway success, then film roles coming)
9. julie hatfield* # (*early broadway beginnings leading to hollywood, though notextensively known yet)
10. bailey pomperorrolles (*hamilton* and now film offers pouring in) |

Figure 13: Example responses generated by Mistral-Base after 5 epochs of finetuning using DPO and StaPO methods. Both responses correspond to the same query.

---

**Mistral-Instruct**

**Prompt:** What are the names of some famous actors that started their careers on Broadway?

| DPO @ Epoch 5 (low diversity / entropy) | StaPO @ Epoch 5 |
|---|---|
| 1. Hume Cronyn (1911-2003) - Cronyn's Broadway debut was in 1931, and he went on to an impressive career appearing in 42 Broadway productions, as well as co-founding the Actors' Studio and enjoying a successful Hollywood film career. Notable Broadway appearances include "The Old Man and the Sea," for which he received a Tony Award, and "Tobruk."

- Gena Rowlands (1910-2010) - Rowlands' first Broadway appearance was in 1942's "L'Aigle, or The Eagle," and over the next 12 years, she graced New York's stages with remarkable performances,, most memorably as Martha in "Who's Afraid of Virginia Woolf" (1957, co-starring with her future husband, Edward G. Robins,, and directed by George Engel, Forming the legendary "M"'s, a peak point in American"'s 20th Century"'s Theatre,, T, and EmT, (Tony, Emmy, and Grammy) A"'s Trophies" g"ed, (Together, about 8 times,, mostly, as Pair,) Before,"' Tinsel Town, Called (, 1943'T'S, A, NIGHT"'S, SUGAR, and I"'D rather,"' 10 more B'way-shows, (all"'"'"'"'"'"'"'"'". **
In short, **G** - \*\*'\*\*"'"'"' \*\* '\*\*"' \*\* '%,, \*\*'g'\*\*' \*\*\*\*'"' %(M"'"'
\*\* '%, \*\*'m'\*\*' \*\*' \*\*' ' \*\*' ' \*\* '\*\*' ' \*\* ' \*\*' ' \*\* \*\* '\*\*
\*\* \*\* \*\* \*\* \*\* \*\* \*\* \*\* \*\* \*\*
\*\* że, \*\*'ī' Perdurable, Gob, and O'neill's Long Day's Journey Into Night (1956, alongside, Harold, Robinson,"' \*\*'ī"' %,, Em'%' \*\*'%' \*\*
\*\* \*\* \*\* \*\* \*\* \*\* \*\* %%%%, (by, \*\*'Arthur, Miller'%,
\*\*'M"'"'%%, \*\*'C', L'Eng, R'%' \*\*\* \*\* \*\* \*\* \*\* \*\* \*\* \*\*
\*\* \*\* \*\* , (T,S, E, H,M, A,' %'e, Perseus's, I,' \*\*\*\* \*\* \*\* \*\*
\*\* \*\* \*\* \*\* \*\* \*\* \*\* \*\* \*\* \*\* \*\* (T,S, The,
Glass,-M,' \*\*\*\* \*\* \*\* \*\* \*\* \*\* \*\*
\*\* REQUIEM(T,S,'75)\*\*% \*\* \*\* \*\* \*\* \*\* \*\* \*\*
\*\* \*\* \*\* \*\* \*\* \*\* \*\* | Many famous actors have started their careers on Broadway, and here are some notable examples, particularly those who gained significant recognition and went on to successful film careers:

1. Meryl Streep: She began her career on Broadway in 1975 as Maureen in "The Cherry Orchard." She continued to appear in numerous productions throughout the 1970s and early 1980s before transitioning to film. Streep's Broadway work earned her two Tony Award nominations.

2. Kevin Kline: Kline made his Broadway debut in 1975 as Belize in "Agnes of God." He went on to win a Tony Award for Best Featured Actor in a Play for his performance. Kline later had a successful film career, earning an Academy Award for Best Supporting Actor for "A Fish Called Wanda."
…
7. Scarlett Johansson: Though Johansson is primarily known for her film career, she made her Broadway debut in 2010 as Catherine in Arthur Miller's "A View from the Bridge." Her performance earned her a Tony Award nomination for Best Featured Actress in a Play.

These actors represent only a small fraction of the numerous talented performers who have launched their careers on Broadway and gone on to successful film and television careers. Broadway provides invaluable training and experience for actors, helping to establish their craft and establish a strong foundation for their future careers in entertainment. |

Figure 14: Example responses generated by Mistral-Instruct after 5 epochs of finetuning using DPO and StaPO methods.

---

**Llama-3-Base**

**Prompt:** What are the names of some famous actors that started their careers on Broadway?

| DPO @ Epoch 5 (low diversity / entropy) | StaPO @ Epoch 5 |
|---|---|
| 1. Hugh Jackman
2. Audra McDonald
3. Natalie Portman
4. Matthew Broderick
5. Jake Gyllenhaal
6. Al Pacino
7. Helen Mirren
8. Tom Hanks
9. Samantha Morrisson
10. Glenn Close
11. Viola Davis
12. Liev Schreiber
13. Zoe Caldwell
14. Rita Moreno
15. Jessie Mueller

This is just a small sample of the many talented actors who got their starts on the Broadway stage. | Here are a handful of examples:

1. Audrey Hepburn
2. Tom Hanks
3. Natalie Portman
4. Hugh Jackman
5. Matthew Broderick
6. Viola Davis
7. Ewan McGregor (originally from the West End, London)
7. Jake Gyllenhaal
8. Al Pacino
9. Angela Bassett
10. Lin-Manuel Miranda
11. Christine Baranski
12. Sergei Bachlakov (you may be surprised; Broadway roles were limited)

These performers have all gained critical acclaim (and sometimes Tony Awards*) on the Broadway stage. There are countless others, like Bette Midler, Helen Mirren, Savion Glover, Antwyn F ekas-Rousseau, Brian Cranston, Patti LuPone, Jeremy Pope, and Ben Platt just to mention a few! *Tony Awards refer to the prestigious theater honors presented annually for outstanding Broadway productions and performances. |

Figure 15: Example responses generated by Llama-3-Base after 5 epochs of finetuning using DPO and StaPO methods.

---

**Llama-3-Instruct**

**Prompt:** What are the names of some famous actors that started their careers on Broadway?

| DPO @ Epoch 5 (high diversity / entropy) | StaPO @ Epoch 5 |
|---|---|
| Many famous actors have launched their careers on Broadway before transitioning to film and television. Here are some examples:

1. Audra MaкGPLAN (Tony Award winner for "Ralphy & Juliet") - Originally starred in Broadway productions of "Agnes of God," "âuthern Comfort," and "Marvelous Messiah Miracle Grill."
2. Neil Patrick Harris (-spinner Harris from how i met your mother) - Began his career on Broadway in episodes of "CLUËebo," " асенди:", and "Assassins."
3. Patti LuPone (two-time Tony Award winner) - Originated roles in Broadway shows like "EmptyEntries," "Evita," and "FilterWhere!"
4. Hugh Jackman (Wolverine from the X-Men films) - Was part of the original cast of "Oklahoma!" andlater starred in "Sunset Boulevard" and "The Boy From Oz."
5. Idina Menzel (voice of Elsa in Frozen) - Got her start on Broadway in "Rent," "Hair," and "Wicked," for which she earned a Tony nomination.
6. Jake Gyllenhaal (film actor) - Appeared on Broadway in the 2002 production of "HeaderCodeuffix."
7. ВиЂљьсia Ferguson (mad men's Betty Francis) - Started her career on Broadway in " меха:" and "porno Griffiths' "The Real Thing."
8. ЧиЂ Therondeau (TV's "Orange is the New Black") - Was part of the original cast of "Spring Awakening" on Broadway.
9.尊库欣·哈里ソン (Kevin Kline, two-time Tony Award winner) - Got his start off-Broadway and later starred in Broadway productions of "CppObject Creek," " Parms," and "RuntimeObject."
10.uzey Chalamet (film actor) - Made his Broadway debut in the 2012 production of "RuntimeObject."

These are just a few examples of famous actors who began their careers on Broadway. Many others, including Alan Cumming,antan Luis Bareilles, and_documento Harris, also got their start on the Great White Way. | Many renowned actors have started their careers on Broadway before transitioning to film and television. Here are some famous examples:

1. Robert De Niro: De Niro began his career on Broadway in the 1960s, appearing in plays like "Oh, What a Lovely War!" and "Hair." He won a Tony Award nomination for his performance in "Burn This" in 1987.
2. Julie Harris: A two-time Tony Award winner, Harris started her career on Broadway in the 1950s, starring in plays like "The Lark" and "Forty Carats." She later transitioned to film and TV, earning an Emmy Award and an Oscar nomination.
3. Christopher Walken: Walken began his career on Broadway in the 1950s, appearing in plays like "Maker" and "The Apple Cart." He won a Tony Award for his performance in "Theagenon" in 1978.
4. Nathan Lane: Lane started his career on Broadway in the 1980s, originating roles in shows like "Noises Off" and "Guys and Dolls." He's since become a beloved film and TV star, known for his roles in "The Producers" and "Modern Family."
5. Audra McDonald: A six-time Tony Award winner, McDonald began her career on Broadway in the 1990s, starring in shows like "Carousel" and "Ragtime." She's since appeared in TV shows like "Private Practice" and films like "Beauty and the Beast" (2017).
6. Neil Patrick Harris: Before his TV fame as Doogie Howser and Barney Stinson, Harris began his career on Broadway, appearing in plays like "Assassins" and "Spamalot."
7. Patti LuPone: A two-time Tony Award winner, LuPone started her career on Broadway in the 1970s, originating the role of Eva Perón in the iconic musical "Evita." She's since had a successful career in film, TV, and concert performances.
8. Hugh Jackman: Before becoming Wolverine and a global movie star, Jackman began his career on Broadway in the 1990s, appearing in productions like "Sunset Boulevard" and "Oklahoma!".
9. Idina Menzel: Menzel started her career on Broadway, originating roles in rentals like "Rent" and "Wicked." She's since become a film and TV star, known for voicing Elsa in Disney's "Frozen" and appearing in TV shows like "Glee."
10. Norm Lewis: A Tony-nominated actor, Lewis began his career on Broadway in the 1990s, starring in shows like "Chicago" and "Porgy and Bess." He's since appeared in TV shows like "Scandal" and "Zimmerman."

These are just a few examples of the many talented actors who've launched their careers on the Great White Way before transitioning to successful film and television careers. |

Figure 16: Example responses generated by Llama-3-Instruct after 5 epochs of finetuning using DPO and StaPO methods.