# OpenReview forum: "Stable Preference Optimization via Two-sided Contrastive Learning"
_ICLR.cc/2026/Conference — Submitted to ICLR 2026_

### Official Review · Reviewer_Ppu7 · 2025-10-29

**Soundness:** 2
**Presentation:** 3
**Contribution:** 3
**Rating:** 4
**Confidence:** 4

**Summary:**

This paper proposes Stable Preference Optimization (StaPO), which introduces a two-sided contrastive objective to stabilize offline preference optimization. Instead of unboundedly enlarging the logit gap between preferred and dispreferred responses (as in DPO/SimPO), StaPO constrains the gap within a bounded interval using left and right margins. Experiments on UltraFeedback with Mistral-7B and Llama-3-8B show that StaPO yields more stable training dynamics and consistent improvements on AlpacaEval 2.0, Arena-Hard, and MT-Bench.

**Strengths:**

1. The proposed idea is simple yet effective. Adding a right-margin constraint to prevent over-optimization is intuitive and yields clear empirical benefits.

2. The writing is clear and well structured, which makes the technical idea easy to follow.

**Weaknesses:**

1. Based on my experience with DPO-style training, the probabilities of both chosen and rejected responses often decrease simultaneously, rather than the logit gap increasing monotonically as claimed. Including training-curve plots of chosen vs. rejected probabilities would make the argument more convincing.

2. StaPO requires tuning two margin hyperparameters (mₗ, mᵣ), adding complexity compared to SimPO or DPO.

3. In Table 3, the main results are reported at epoch 5, while DPO and SimPO typically overfit in later epochs. A fairer comparison would evaluate baselines at their best epoch (e.g., epoch 2–3). After several epochs, these methods often produce longer responses and show performance drops on knowledge-intensive QA tasks (e.g., OpenLLM leaderboard). Including such indicators would better demonstrate StaPO’s resistance to overfitting.

**Questions:**

Please see the “Weaknesses” section above.

---

> ### Author Response · Authors · 2025-12-02
> **Response to Reviewer Ppu7 (Part 1)**
>
> We appreciate your constructive feedback. Below, we address your concerns.
>
> ---
>
> **Comment 1: Based on my experience with DPO-style training, the probabilities of both chosen and rejected responses often decrease simultaneously, rather than the logit gap increasing monotonically as claimed. Including training-curve plots of chosen vs. rejected probabilities would make the argument more convincing.**
>
> **Response1**: We thank the reviewer for the insightful comment.
>
> In the revised paper, we have added training-curve plots of the chosen and rejected response probabilities (see Figure 3). The log probabilities are averaged over the mini-batches of the training set.
>
> As observed, the probabilities of both responses indeed decrease during training; however, **this does NOT contradict our claim about the increasing logit gap**. The key observation is that **the probability of the rejected response decreases at a faster rate**, leading to a monotonically widening logit gap.
>
> For DPO and SimPO, this gap continues to increase throughout finetuning, as neither method imposes explicit regularization on the contrastive logit. In contrast, StaPO’s logit gap initially rises from 0 to 0.6 and then converges around 0.63, which aligns well with our margin configuration $m_l = 0.6$ and $m_r = 0.8$. This consistent behavior across Figures 2 and 3 further supports our explanation that StaPO effectively stabilizes contrastive learning dynamics.
>
> &nbsp;
>
> **Comment 2: StaPO requires tuning two margin hyperparameters (mₗ, mᵣ), adding complexity compared to SimPO or DPO.**
>
> **Response 2**: In this paper, we adopt a simple but effective margin tuning strategy: we fix the margin gap ($m_r - m_l$) and jointly adjust both $m_l$ and $m_r$. **This strategy reduces the tuning process to essentially one hyperparameter.**
>
> In our experiments, a margin gap of $m_r - m_l = 0.2$ consistently yielded strong results across all datasets. Additionally, StaPO exhibits reduced sensitivity to the number of finetuning epochs, further minimizing the tuning effort required in practice.
>
> Alternatively, one could fix $m_l$ and gradually decrease $m_r$ from a larger value toward $m_l$. This flexibility in tuning is enabled by the clear physical meaning of the two margins: $m_l$ governs the effectiveness of preference learning, while $m_r$ controls training stability. Theoretically, $m_r$ can range from $m_l$ to $+\infty$, with values closer to $m_l$ providing greater stability.
>
> We also provide ablation studies in Table 1-2 to further demonstrate that StaPO’s performance is robust to margin choices, maintaining stable results across a wide range of margins and finetuning epochs.
>
> >Table 1: Varying $m_l$ and $m_r$ for StaPO method with “fixed-gap” strategy
> >| $m_l$ | $m_r$ | Epoch 1 | Epoch 3 | Epoch 5 |
> >|---------|-----------|------------|--------------|---------|
> >|  0.4  |  0.6  |  57.88    |   59.62   |   60.46   |
> >|  0.6  |  0.8  |  58.84    |   61.22   |   **61.97**   |
> >|  0.8  |  1.0  |  59.50    |   61.15   |   61.49   |
>
>
> >Table 2: Varying $m_l$ and $m_r$ for StaPO method with “right-search” strategy
> >| $m_l$ | $m_r$ | Epoch 1 | Epoch 3 | Epoch 5 |
> >|---------|-----------|------------|--------------|---------|
> >|  0.4  |  0.8  |  57.29  |  59.45  |  61.20  |
> >|  0.4  |  1.0  |  57.03  |  60.03  |  60.84  |
> >|  0.6  |  1.0  |  59.04  |  60.84  |  **61.33**  |
> >|  0.6  |  1.2  |  60.17  |  59.32  |  58.54  |

---

> ### Author Response · Authors · 2025-12-02
> **Response to Reviewer Ppu7 (Part 2)**
>
> **Comment 3: In Table 5, the main results are reported at epoch 5, while DPO and SimPO typically overfit in later epochs. A fairer comparison would evaluate baselines at their best epoch (e.g., epoch 2–3).**
>
> **Response 3**: There is a **misunderstanding**: the results in Table 5 are NOT reported solely at epoch 5.
> We evaluate every model across all finetuning epochs (1–5) and report **each method’s best-performing epoch**. Thus, the actual epoch number may vary between methods depending on when they achieve their peak performance.
>
> This ensures a fair and representative comparison. To make this clearer, we have revised the table caption and added a note in the main text specifying that all results correspond to the best epoch per method.
>
> &nbsp;
>
> **Comment 4: After several epochs, these methods often produce longer responses and show performance drops on knowledge-intensive QA tasks (e.g., OpenLLM leaderboard). Including such indicators would better demonstrate StaPO’s resistance to overfitting.**
>
> **Response 4**: Indeed, we observe that models finetuned with existing OsC methods tend to produce longer responses after several epochs, which often leads to degraded performance on knowledge-intensive QA tasks. StaPO, by contrast, substantially mitigates this issue through its dual-margin constraint, which stabilizes optimization and helps preserve the base model’s knowledge.
>
> To evaluate this effect, we conducted experiments on representative knowledge-intensive benchmarks from the OpenLLM Leaderboard, including ARC-Challenge, MedMCQA, PIQA, and WebQS, using the official lm-evaluation-harness framework (https://github.com/EleutherAI/lm-evaluation-harness). As shown in the results, **SimPO exhibits notable performance degradation with continued finetuning, whereas StaPO demonstrates slower forgetting and, in some cases, even improvement**: for instance, StaPO shows drops on MedMCQA (60.70 to 57.69 vs. 60.70 to 56.83 for SimPO) and PIQA (78.02 to 74.97 vs. 78.02 to 65.61 for SimPO), and achieves slight gains on ARC-Challenge (53.24 to 54.18) and WebQS (5.61 to 10.53).
>
> These results confirm that StaPO effectively resists overfitting while maintaining factual knowledge and generalization performance across diverse QA tasks.
>
> >Table 3: Comparison of SimPO and StaPO across epochs on knowledge-intensive QA tasks
> >| Method | # Epoch | ARC-Challenge | MedMCQA | PIQA | WebQS |
> >|---------|:----------:|:----------:|:----------:|:----------:|:----------:|
> >| Llama-3-Instruct (8B) |   0  | 53.24 | **60.70** | **78.02** | 5.61 |
> >|  SimPO  |  1  | 51.96 | 58.36 | 72.74 |   6.15 |
> >|  SimPO  |  3  | 49.23 | 56.66 | 67.63 | 10.33 |
> >|  SimPO  |  5  | 45.90 | 56.83 | 65.61 |   7.97 |
> >|  StaPO   |  1  | **55.55** | 59.36 | 76.71 |   6.10 |
> >|  StaPO   |  3  | 54.78 | 58.12 | 74.65 | 10.48 |
> >|  StaPO   |  5  | 54.18 | 57.69 | 74.97 | **10.53** |

---

### Official Review · Reviewer_raBq · 2025-10-31

**Soundness:** 3
**Presentation:** 3
**Contribution:** 3
**Rating:** 6
**Confidence:** 3

**Summary:**

This paper unifies existing offline preference optimization methods under a one-sided contrastive (OsC) learning framework and highlights that unconstrained maximization of the contrastive logit can gradually erode the LLM’s core linguistic capabilities. To address this, the authors propose StaPO, a two-sided contrastive (TsC) learning framework that both facilitates preference learning and constrains the excessive growth of contrastive logits.

**Strengths:**

- Provides an insightful unification of current offline preference optimization approaches.

- The two-sided contrastive design is conceptually elegant, and directly addresses a clear limitation of existing OsC-based methods.

- The paper is generally well-written, clearly organized, and easy to follow.

**Weaknesses:**

- Although StaPO claims reduced sensitivity, the margin values (m_l, m_r) still require tuning (or the gap); providing a principled criterion for selecting them would enhance reproducibility.

- The contribution could be strengthened by discussing StaPO’s behavior on out-of-domain or unseen preference distributions (i.e., its potential alignment tax).

- The lack of statistical significance analysis makes it difficult to assess the reliability of the reported improvements.

**Questions:**

1. What is the expression or definition of Z used in StaPO? Have you tested different Z variants listed in the table within the StaPO framework?
2. In Table 2, does this configuration generalize across all model families? Do you have any suggestions regarding the tuning of m_l?
3. How sensitive is the performance to the specific margin gap (m_r - m_l)?

---

> ### Author Response · Authors · 2025-11-28
> **Response to Reviewer raBq (Part 1)**
>
> We appreciate your constructive feedback and insightful questions. Below, we address your concerns one by one.
>
> ---
>
> **Comment 1: Although StaPO claims reduced sensitivity, the margin values (m_l, m_r) still require tuning (or the gap); providing a principled criterion for selecting them would enhance reproducibility. Do you have any suggestions regarding the tuning of m_l? How sensitive is the performance to the specific margin gap (m_r - m_l)?**
>
> **Response 1**: Thank you for the question.
>
> In this paper, we adopt a simple but effective margin tuning strategy **(line 235-239 in the main paper)**: we fix the margin gap ($m_r - m_l$) and jointly adjust both $m_l$ and $m_r$. This strategy reduces the tuning process to essentially one hyperparameter. In our experiments, a margin gap of $m_r - m_l = 0.2$ consistently yielded strong results across all datasets. Additionally, StaPO exhibits reduced sensitivity to the number of finetuning epochs, further minimizing the tuning effort required in practice.
>
> Alternatively, one could fix $m_l$ and gradually decrease $m_r$ from a larger value toward $m_l$. This flexibility in tuning is enabled by the clear physical meaning of the two margins: $m_l$ governs the effectiveness of preference learning, while $m_r$ controls training stability. Theoretically, $m_r$ can range from $m_l$ to $+\infty$, with values closer to $m_l$ providing greater stability.
>
> We also provide ablation studies in Table 1-2 to further demonstrate that StaPO’s performance is robust to margin choices, maintaining stable results across a wide range of margins and finetuning epochs. These results are also included in the revised version of the main paper **(line 348-358)**.
>
> >Table 1: Varying $m_l$ and $m_r$ for StaPO method with “fixed-gap” strategy
> >| $m_l$ | $m_r$ | Epoch 1 | Epoch 3 | Epoch 5 |
> >|:---------:|:---------:|:---------:|:---------:|:---------:|
> >|  0.4  |  0.6  |  57.88    |   59.62   |   60.46   |
> >|  0.6  |  0.8  |  58.84    | **61.22**| **61.97** |
> >|  0.8  |  1.0  |**59.50** |   61.15   |   61.49   |
>
> >Table 2: Varying $m_l$ and $m_r$ for StaPO method with “right-search” strategy
> >| $m_l$ | $m_r$ | Epoch 1 | Epoch 3 | Epoch 5 |
> >|:---------:|:---------:|:---------:|:---------:|:---------:|
> >|  0.4  |  0.8  |  57.29  |  59.45  |  61.20  |
> >|  0.4  |  1.0  |  57.03  |  60.03  |  60.84  |
> >|  0.6  |  1.0  |  59.04  |**60.84**|**61.33**|
> >|  0.6  |  1.2  |**60.17**|  59.32  |  58.54  |
>
> &nbsp;
>
> **Comment 2: The contribution could be strengthened by discussing StaPO’s behavior on out-of-domain or unseen preference distributions (i.e., its potential alignment tax).**
>
> **Response 2**: Thank you for the suggestion.
>
> To evaluate StaPO on out-of-domain or unseen preference distributions, we conducted additional experiments on the Open LLM Leaderboard using [the official lm-evaluation-harness framework](https://github.com/EleutherAI/lm-evaluation-harness). We included diverse benchmarks (ARC-Challenge, MedMCQA, PIQA, and WebQS), covering a range of knowledge-intensive, reasoning, and problem-solving tasks.
>
> As summarized in Table 3, SimPO exhibits notable performance degradation with extended finetuning, while StaPO demonstrates slower forgetting and, in several cases, even improvement. Specifically, StaPO shows smaller drops on MedMCQA (60.70 to 57.69 vs. 60.70 to 56.83 for SimPO) and PIQA (78.02 to 74.97 vs. 78.02 to 65.61 for SimPO), and achieves modest gains on ARC-Challenge (53.24 to 54.18) and WebQS (5.61 to 10.53).
>
> These results indicate that StaPO generalizes more effectively to out-of-domain tasks, exhibiting a lower alignment tax than SimPO while maintaining better performance on in-domain benchmarks. This suggests that StaPO’s dual-margin formulation not only stabilizes finetuning but also preserves generalization across diverse task distributions.
>
> >Table 3: Comparison of SimPO and StaPO across epochs on knowledge-intensive QA tasks
> >| Method | # Epoch | ARC-Challenge | MedMCQA | PIQA | WebQS |
> >|:----------:|:----------:|:----------:|:----------:|:----------:|:----------:|
> >| Llama-3-Instruct (8B) |   0  | 53.24 |**60.70**|**78.02**| 5.61 |
> >|  SimPO  |  1  | 51.96 | 58.36 | 72.74 |   6.15 |
> >|  SimPO  |  3  | 49.23 | 56.66 | 67.63 | 10.33 |
> >|  SimPO  |  5  | 45.90 | 56.83 | 65.61 |   7.97 |
> >|  StaPO   |  1  |**55.55**| 59.36 | 76.71 |   6.10 |
> >|  StaPO   |  3  | 54.78 | 58.12 | 74.65 | 10.48 |
> >|  StaPO   |  5  | 54.18 | 57.69 | 74.97 |**10.53**|

---

> ### Author Response · Authors · 2025-11-28
> **Response to Reviewer raBq (Part 2)**
>
> **Comment 3: The lack of statistical significance analysis makes it difficult to assess the reliability of the reported improvements.**
>
> **Response 3**: Thank you for the valuable comment.
>
> We have reported the average performance across 5 independent finetuning runs for each method. To better demonstrate the statistical significance, we have now **included the standard deviations** in the revised version of Table 5 of the main paper **(line 502,503,515,516)**. All experiments were conducted with different random seeds under identical data splits and hyperparameter settings to ensure fair and reproducible evaluation.
>
> The observed standard deviations are consistently small, confirming that the performance improvements achieved by StaPO are not due to random variation.
>
> &nbsp;
>
> **Comment 4: What is the expression or definition of Z used in StaPO? Have you tested different Z variants listed in the table within the StaPO framework?**
>
> **Response 4**: Thank you for raising this question.
>
> In StaPO, the expression of $Z$ follows the formulation used in SimPO, defined as
> $$Z = \frac{1}{|y_p|}\log \pi_\theta(y_p \mid x) - \frac{1}{|y_n|}\log \pi_\theta(y_n \mid x).$$
>
> We adopt this instantiation because SimPO consistently demonstrates the strongest empirical performance among existing offline preference optimization methods, making it a natural baseline for our framework.
>
> As suggested, we also conducted additional experiments using the $Z$ formulation from DPO, defined as
> $$Z = \log \frac{\pi_\theta(y_p \mid x)}{\pi_{\text{ref}}(y_p \mid x)} - \log \frac{\pi_\theta(y_n \mid x)}{\pi_{\text{ref}}(y_n \mid x)}.$$
>
> Table 4 and 5 show the results on AlpacaEval2 **(line 363-376 in the revised paper)**. We observe that **when equipped with the proposed dual-margin constraint, optimizing the DPO-style contrastive logit leads to more stable training and improved overall performance compared to the original DPO**. It is also worth noting that the DPO-style formulation requires substantially higher computational resources, as it must load the reference model into GPU memory and compute additional reference probabilities during training.
>
> >Table 4: Experiments of different $Z$s with Mistral-Instruct (7B)
> >| method | Epoch 1 | Epoch 3 | Epoch 5 |
> >|------------|------------|--------------|---------|
> >|  DPO  | 45.88 | 11.19  | 9.51 |
> >|  StaPO (Z from DPO) |  45.62 | 46.28  | 46.53 |
> >|  SimPO | 48.93 | 1.54 | 0.74 |
> >|  StaPO (Z from SimPO) | 47.83 | **51.14** | 47.49 |
>
>
> >Table 5: Experiments of different $Z$s with Llama-3-Instruct (8B)
> >| method | Epoch 1 | Epoch 3 | Epoch 5 |
> >|------------|------------|--------------|---------|
> >|  DPO | 51.98 | 55.88 | 49.77 |
> >|  StaPO (Z from DPO)  | 51.43 | 56.14 | 57.93 |
> >|  SimPO | 59.04 | 49.56 | 39.08 |
> >|  StaPO (Z from SimPO) | 58.66 | 60.83 | **61.61** |

---

### Official Review · Reviewer_w7HW · 2025-11-01

**Soundness:** 3
**Presentation:** 3
**Contribution:** 3
**Rating:** 6
**Confidence:** 3

**Summary:**

Existing offline preference optimization methods often cause LLMs to generate nonsensical language over extended fine-tuning. This paper argues this "objective misalignment" stems from the unconstrained maximization of the log-probability difference between preferred and dispreferred responses. It proposes Stable Preference Optimization (StaPO), a novel two-sided contrastive learning framework with dual-margin constraints. While a left margin ensures effective preference learning, a crucial right margin limits excessive logit growth, preventing the model’s linguistic collapse. On benchmarks like AlpacaEval2, StaPO demonstrates significant and stable performance improvements over existing methods, maintaining consistent win rates and avoiding performance degradation across models.

**Strengths:**

- Identifies a Critical Limitation: The paper explicitly defines "objective misalignment"—a core flaw in existing offline preference methods (e.g., DPO, SimPO) where extended fine-tuning leads to degenerate language (nonsensical tokens, incoherent phrases). This addresses a practical pain point often overlooked in prior work.

- Unifies Existing Methods Under a Theoretical Framework: It formalizes DPO, SimPO, CPO, and R-DPO into a one-sided contrastive (OsC) learning framework, showing that all these methods inherently maximize the "contrastive logit" (log-probability difference between preferred/dispreferred responses) without constraints. This unification simplifies understanding of why OsC methods erode LLMs’ linguistic capabilities over time.

- Innovative Two-Sided Contrastive (TsC) Design: The proposed StaPO introduces dual-margin constraints (left margin for preference alignment, right margin to limit excessive contrastive logits) to balance preference learning and linguistic coherence. The gradient analysis (Section 3.2) clearly explains how TsC stabilizes training—e.g., gradients adjust bidirectionally to keep logits within bounds, preventing deterministic collapse.

**Weaknesses:**

- While the paper mentions fixing the margin gap (mᵣ - mₗ = 0.2) works well across datasets, it provides no justification for why 0.2 is optimal (e.g., no ablation on gap sizes other than 0.2). The right margin’s impact on "linguistic coherence" is measured indirectly via entropy, but there is no qualitative analysis of how different margin values affect specific linguistic traits (e.g., fluency, factuality, or diversity in open-ended generation).
- The paper illustrates degenerate outputs with 2 examples (Figures 4–5: low-entropy repetitive text, high-entropy mixed-language nonsense), but it lacks a systematic categorization of degenerate patterns (e.g., token repetition, topic drift, logical inconsistency) or quantitative metrics for coherence (e.g., BLEU for fluency, F1 for factuality). It does not test whether StaPO avoids other common failures of OsC methods (e.g., "reward hacking"—overfitting to preference data at the cost of generalizability).
- The paper focuses on comparing StaPO to OsC methods (DPO, SimPO, etc.) but largely ignores non-contrastive offline methods like IPO (Azar et al., 2024) or RRHF (Yuan et al., 2023) in depth.

**Questions:**

see weakness

---

> ### Author Response · Authors · 2025-12-02
> **Response to Reviewer w7HW (Part 1)**
>
> We thank the reviewer for the questions. Below, we address your concerns one by one.
>
> ---
>
> **Comment 1: While the paper mentions fixing the margin gap (mᵣ - mₗ = 0.2) works well across datasets, it provides no justification for why 0.2 is optimal (e.g., no ablation on gap sizes other than 0.2).**
>
> **Response 1**: Good question!
>
> The margin gap $m_r - m_l$ was determined through empirical hyperparameter search. As described in the main paper **(lines 243–247)**, we adopt **a simple yet effective strategy**: the margin gap is fixed and both $m_l$ and $m_r$ are adjusted jointly, which reduces the tuning process to a single hyperparameter.
>
> We conducted additional ablations with larger margin gaps (0.4 and 0.6). The results, summarized in Table 1 (also included in the revised version of the main paper, **line 348-358**), show that $m_r - m_l = 0.2$ achieves the best trade-off between alignment effectiveness and stability. These experiments provide strong empirical justification for its selection.
>
> >Table 1: Varying margin gap $m_r - m_l$ for StaPO method
> >|$m_r - m_l$ | $m_l$ | $m_r$ | Epoch 1 | Epoch 3 | Epoch 5 |
> >|:------:|:------:|:-------------:|:---------:|:---------:|:---------:|
> >|  0.2  |  0.2  |  0.4  |  54.92  |  58.14  |  59.63  |
> >|  0.2  |  0.4  |  0.6  |  57.88  |  59.62  |  60.46  |
> >|  0.2  |  0.6  |  0.8  |  58.84  |**61.22**|**61.97**|
> >|  0.2  |  0.8  |  1.0  |  59.50  |  61.15  |  61.49  |
> >|  0.4  |  0.4  |  0.8  |  57.29  |  59.45  |  61.20  |
> >|  0.4  |  0.6  |  1.0  |  59.04  |  60.84  |  61.33  |
> >|  0.6  |  0.4  |  1.0  |  57.03  |  60.03  |  60.84  |
> >|  0.6  |  0.6  |  1.2  |**60.17**|  59.32  |  58.54  |
>
> &nbsp;
>
> **Comment 2: The right margin’s impact on "linguistic coherence" is measured indirectly via entropy, but there is no qualitative analysis of how different margin values affect specific linguistic traits (e.g., fluency, factuality, or diversity in open-ended generation).**
>
> **Response 2**: We thank the reviewer for this thoughtful suggestion.
>
> To evaluate the effect of the right margin on linguistic coherence, we employ multiple complementary metrics targeting fluency, factuality, and diversity. Specifically, we use Perplexity and LLM scoring for fluency, the accuracies on ARC-Challenge and PIQA benchmarks for factuality, and Distinct-n for diversity. Lower perplexity and higher LLM scores indicate more fluent text, while higher factual benchmark scores reflect stronger factual knowledge. For Distinct-n, a stable value across epochs suggests consistent lexical diversity rather than uncontrolled randomness.
>
> We compare SimPO, which can be viewed as a variant of StaPO with an infinite right margin, against our StaPO. As shown in Table 2 (**also included in the Appendix, line 906-917**), SimPO leads to degraded fluency and factual accuracy while exhibiting unstable increases in diversity as finetuning progresses. In contrast, StaPO (with proper right-margin constraint) maintains balanced and coherent generation across all three aspects: **it preserves fluency, sustains factual consistency, and stabilizes diversity**. These results demonstrate that the right margin directly contributes to maintaining linguistic coherence in a holistic sense, not just as measured by entropy.
>
> >Table 2: The effect of the right margin $m_r$ on linguistic coherence.
> >| Method | $m_r$ | # Epoch | Perplexity (↓) | LLM scoring (↑) | ARC Challenge (↑) | PIQA (↑) | Distinct-1 | Distinct-2 |
> >|:---------:|:-----------:|:------------:|:--------------:|:---------:|:---------:|:---------:|:---------:|:---------:|
> >|SimPO  | $\infty$ |  1 | 3.004 | 8.149  | 0.519 | 0.727 | 0.555 | 0.854 |
> >|SimPO  | $\infty$ |  3 | 7.257 | 7.885  | 0.492 | 0.676 | 0.597 | 0.895 |
> >|SimPO  | $\infty$ |  5 | 11.290 | 7.675  | 0.459 | 0.656 | 0.618 | 0.905 |
> >|StaPO  | $0.8$     |  1 | **2.826** | 8.156  | **0.555** | **0.767** | 0.539 | 0.842 |
> >|StaPO  | $0.8$     |  3 | 3.022 | **8.169**  | 0.547 | 0.746 | 0.538 | 0.852 |
> >|StaPO  | $0.8$     |  5 | 2.915 | 8.137  | 0.541 | 0.749 | 0.535 | 0.849 |

---

> ### Author Response · Authors · 2025-12-02
> **Response to Reviewer w7HW (Part 2)**
>
> **Comment 3: The paper illustrates degenerate outputs with 2 examples (Figures 4–5: low-entropy repetitive text, high-entropy mixed-language nonsense), but it lacks a systematic categorization of degenerate patterns (e.g., token repetition, topic drift, logical inconsistency) or quantitative metrics for coherence (e.g., BLEU for fluency, F1 for factuality). It does not test whether StaPO avoids other common failures of OsC methods (e.g., "reward hacking"—overfitting to preference data at the cost of generalizability).**
>
> **Response 3**: We appreciate the insightful comment.
>
> We have added more qualitative examples of degenerate outputs in the appendix for better illustration. As observed, degenerate responses often exhibit mixed characteristics, such as token repetition, incoherent topic shifts, and partial logical collapse. Making a strict categorization difficult in practice.
>
> Regarding quantitative evaluation, we note that BLEU and F1 are not suitable for assessing fluency and factuality in open-ended generation. Instead, we employ more reliable measures: Perplexity and LLM-based scoring for fluency, ARC-Challenge and PIQA for factuality, and Distinct-n for lexical diversity. The results, shown in Table 2 from **Response 2** (**also included in the Appendix**), demonstrate that StaPO maintains high fluency and factual consistency while preserving stable diversity across epochs. In contrast, SimPO exhibits reduced fluency and factual accuracy with unstable diversity.
>
> These results indicate that StaPO effectively mitigates the degeneration observed in one-sided contrastive methods and avoids overfitting to preference data, showing no evidence of “reward hacking” or loss of generalization.
>
> &nbsp;
>
> **Comment 4: The paper focuses on comparing StaPO to OsC methods (DPO, SimPO, etc.) but largely ignores non-contrastive offline methods like IPO (Azar et al., 2024) or RRHF (Yuan et al., 2023) in depth.**
>
> **Response 4**: Thank you for the suggestion.
>
> The original version included direct comparisons between StaPO, IPO, and RRHF in Table 5 of the main paper, and a detailed discussion of IPO in the related work section **(line 119-121)**. Below, we summarize our discussion of RRHF, which has been incorporated into the revised version of the main paper **(line 121-123)** for completeness.
>
> RRHF (Yuan et al., 2023) proposes a ranking-based approach that aligns model outputs with human preferences by comparing log-probabilities of sampled responses. As a non-contrastive offline alignment method, RRHF can be viewed as a lightweight extension of supervised finetuning (SFT), leveraging ranking loss to align model behavior with preference signals.

---

### Official Review · Reviewer_XFgv · 2025-11-03

**Soundness:** 3
**Presentation:** 2
**Contribution:** 2
**Rating:** 4
**Confidence:** 4

**Summary:**

The submission proposes a partial unification of existing preference optimization work into a one-sided contrastive framework and further propose a two-sided contrastive loss that is more fit for language models. Empirical validation is done using two model families and 3 evaluation datasets.

**Strengths:**

- Empirical validation does seem favorable compared to existing works
- The idea of a two-sided contrastive loss with double margins is straightforward

**Weaknesses:**

- The method lacks theoretical grounding. The only non-empirical evidence provided by the authors is the following quote"However, this discriminative principle conflicts with generative
language modeling, which requires maintaining balanced probability distributions across multiple
plausible outputs to ensure linguistic coherence and diversity."
However, this quote has no references, or other more in-depth justification in the given manuscript. Limiting deviation from the original model was exactly what the KL divergence term in the DPo was for, and yet the paper fails to explain why this doesn't work in practice. The given solution is a somewhat hard clip of the contrastive logit value, which seems to lack theoretical justification, albeit effective empirically

**Questions:**

- Does the DPO loss also degenerate like SimPO despite the KL divergence term?

---

> ### Author Response · Authors · 2025-11-28
> **Response to Reviewer XFgv**
>
> Thank you for your detailed review. Below, we address your concerns and suggestions.
>
> ---
>
> **Comment 1: The method lacks theoretical grounding. The only non-empirical evidence provided by the authors is the following quote"However, this discriminative principle conflicts with generative language modeling, which requires maintaining balanced probability distributions across multiple plausible outputs to ensure linguistic coherence and diversity." However, this quote has no references, or other more in-depth justification in the given manuscript.**
>
> **Response1**: Thanks for the valuable suggestions.
>
> We provide **a detailed justification** and their **corresponding reference** below. The reference and a compact version of this justification are also incorporated to the introduction section of the revised main paper **(line 70-74)**.
>
> Generative language modeling requires maintaining balanced probability distributions across multiple plausible outputs to ensure linguistic coherence and diversity, because natural language itself is intrinsically uncertain and multi-modal in its possible trajectories [1]. A well-behaved language model must therefore allocate probability mass in a calibrated manner that reflects these underlying linguistic uncertainties, rather than collapsing prematurely onto a single mode [2]. For this reason, balanced probability distributions are not an optional aesthetic property but a core requirement for generative language modeling to faithfully represent the stochastic structure of language and produce coherent, diverse, and contextually grounded outputs [3, 4]. Our motivation to constrain contrastive logits directly stems from this principle.
>
> > **Reference:**
> >
> > [1] Holtzman A, Buys J, Du L, et al. "The Curious Case of Neural Text Degeneration", ICLR 2020.
> >
> > [2] Lovering, Charles, et al. "Language model probabilities are not calibrated in numeric contexts." ACL 2025.
> >
> > [3] Zhou, Yuxuan, , et al. "Balancing diversity and risk in llm sampling: How to select your method and parameter for open-ended text generation." ACL 2025.
> >
> > [4] Karlgren, Jussi, and Magnus Sahlgren. "Culture, Language, and Generative Language Models." Communications of the ACM (2025).
>
> &nbsp;
>
> **Comment 2: Limiting deviation from the original model was exactly what the KL divergence term in the DPo was for, and yet the paper fails to explain why this doesn't work in practice. The given solution is a somewhat hard clip of the contrastive logit value, which seems to lack theoretical justification, albeit effective empirically.**
>
> **Response 2**: We believe there is a **misunderstanding** regarding the presence of a KL divergence term in DPO.
>
> While DPO is derived from the RLHF framework with a KL penalty term, **the final DPO objective itself does NOT contain a KL regularization component**. As shown in Equation (3) of our paper, DPO optimizes a contrastive logit of the log-probability ratios between the policy and reference models. There is no mechanism in DPO to limit the unconstrained growth of the contrastive logit value, i.e.,
> $$\log \pi_{ref}(y_p | x) - \log \pi_{ref}(y_n | x) < \log \pi_\theta(y_p | x) - \log \pi_\theta(y_n | x).$$
>
> As illustrated in Figure 1(b, left), when this difference grows excessively, the model becomes overconfident in the preferred responses and gradually deviates from the linguistic balance of the base model, a behavior that we empirically confirm in Section 4.
>
> To address this problem, our proposed StaPO explicitly introduces dual-margin constraints on the contrastive logit,
> $$m_l < \log \pi_\theta(y_p | x) - \log \pi_\theta(y_n | x) < m_r,$$
>
> As shown in Figure 1(b, right), the left margin maintains effective preference learning, while the right margin limits excessive logit growth, ensuring that optimization remains within a “healthy” range. This design is analogous to introducing a bounded confidence region in contrastive learning, ensuring that optimization neither underfits nor overfits user preference signals.
>
> &nbsp;
>
> **Comment 3: Does the DPO loss also degenerate like SimPO despite the KL divergence term?**
>
> **Response 3**: Again, this is a **misunderstanding**.
>
> To clarify, **neither DPO nor SimPO includes an explicit KL divergence term** in their final optimization objective.
> Although DPO was originally derived from the RLHF framework with a KL regularization term, the DPO formulation replaces the KL term with a contrastive objective between preferred and dispreferred responses. As a result, there is NO explicit mechanism in DPO or SimPO to prevent the contrastive logit from growing without bound.
>
> Consequently, both DPO and SimPO exhibit similar degeneration behaviors during extended finetuning, as shown in our theoretical illustration (Figure 1b) and empirical results (Figures 2 and 3). Both methods gradually over-optimize the contrastive logit, leading to unstable training dynamics and eventual degradation of linguistic coherence and diversity.

---

### Author Response · Authors · 2025-11-26
**Request for Brief Extension**

Dear Reviewers,

&nbsp;

We sincerely appreciate the time and effort you have spent on our submission. Your comments are valuable in helping us refine and strengthen the paper.

We are currently running several additional experiments to ensure that our responses are complete and supported by solid evidence. However, these experiments are taking slightly longer than expected. To provide you with the most accurate and well-substantiated rebuttal, we kindly request two additional days to finalize our responses.

We apologize for this minor delay and greatly thank your understanding and patience.

&nbsp;

Warm regards,

The Authors

---

### Author Response · Authors · 2025-12-04
**Summary (details in respective responses)**

#### **Reviewer XFgv** recognizes the unified OsC framework and the proposed TsC formulation, StaPO, as a straightforward and effective approach. The empirical results are consistently favorable across model families and benchmark datasets.

#### Other comments:

#### **1. Lack of theoretical grounding and suggestion to include more in-depth justification with references.**

> We added detailed theoretical justification and supporting references, now incorporated into the revision **(Lines 70–74)**.

#### **2. Question on why StaPO is needed when DPO already includes a KL term for deviation control.**

> **This is a misunderstanding**. The DPO objective itself does NOT contain a KL regularization term.

#### **3. Question on whether DPO also degenerates like SimPO despite the KL divergence term.**

> Neither DPO nor SimPO includes an explicit KL term; both exhibit similar degeneration behaviors during extended finetuning, as shown in our analysis **(Fig. 1b)** and empirical results **(Fig. 2 and 3)**.

---

#### **Reviewer w7HW** appreciates the paper’s clear motivation, the unified OsC framework, and the introduction of a TsC design that effectively stabilizes preference optimization. The reviewer also acknowledges the consistent improvements over existing methods.

#### Other comments:

#### **1. Limited analysis of the right-margin’s effect on linguistic coherence.**

> As suggested, we add new experiments **(Lines 906-917)** varying the right margin, evaluating linguistic coherence from fluency (perplexity, LLM scoring), factuality (ARC-Challenge, PIQA), and diversity (Distinct-n).

#### **2. Limited examples of degenerate outputs.**

> More qualitative examples are in the original Appendix. Quantitatively, StaPO maintains linguistic coherence across fluency, factuality, and diversity metrics **(Lines 906-917)**.

#### **3. Missing comparison with non-contrastive baselines (IPO, RRHF).**

> The main paper already includes comparisons with IPO and RRHF (Tab. 5), and discusses IPO **(Lines 119-121)**. We now add a discussion of RRHF **(Lines 121–123)** for completeness.

---

#### **Reviewer D3Qx** considers the unification of existing OsC methods insightful and the proposed TsC design conceptually elegant. The reviewer also appreciates the strong empirical results and well-organized presentation.

#### Other comments:

#### **1. Limited discussion on StaPO’s generalization to out-of-domain or unseen preference distributions (alignment tax).**

> We extend evaluation to out-of-domain tasks on the Open LLM leaderboard. StaPO shows slower forgetting on MedMCQA and PIQA, and performance gains on ARC-Challenge and WebQS, indicating lower alignment tax than SimPO.

#### **2. Lack of statistical significance analysis.**

> We now report **standard deviation** across 5 independent runs in **Tab. 5**, showing small variance and confirming StaPO’s improvements are statistically reliable.

#### **3. Unclear definition of $Z$ in StaPO and evaluation of other $Z$s variants in Tab. 1.**

> We clarify that $Z$ in StaPO follows SimPO’s definition, and also test DPO-style $Z$. When equipped with the dual-margin constraint, both forms yield more stable training and improved performance.

---

#### **Reviewer s7DL** appreciates StaPO’s simple yet effective design and the well-structured presentation that makes the proposed TsC formulation easy to follow. The reviewer also notes that adding a right-margin constraint is an intuitive and empirically effective way to prevent over-optimization.

#### Other comments:

#### **1. Question about the claim that the logit gap increases monotonically, suggesting both chosen and rejected probabilities decrease simultaneously.**

> While both probabilities drop during finetuning, the rejected ones decline faster, yielding a consistently larger logit gap. Supporting training-curve plots are now added in Fig. 3 **(Lines 329–344)**.

#### **2. Question on whether Tab. 5 reports results at epoch 5 or each method’s best epoch.**

> We confirm that Tab. 5 reports results at each method’s best-performing epoch, not necessarily epoch 5, which ensures fair comparison.

#### **3. Suggestion to show StaPO’s robustness on knowledge-intensive QA tasks.**

> Additional experiments on the Open LLM Leaderboard show StaPO exhibits slower forgetting and even improvements **(Lines 379-410)**, confirming robustness to overfitting.

---

#### **Reviewers w7HW, D3Qx, and s7DL raised a common question: StaPO’s dual-margin design lacks a principled justification for its specific values and adds tuning complexity compared to DPO/SimPO.**

> We describe our tuning strategy in the original paper **(Lines 243–247)**, which **reduces tuning to a single hyperparameter** by fixing the margin gap $m_r - m_l$ and jointly adjusting both margins. Extensive ablations across different gaps (0.2, 0.4, 0.6) confirm that the same setting can yield consistent performance across datasets **(Tab. 2, Lines 348–359)**.

---

### Meta-Review · Area_Chair_XuwV · 2026-01-03

**Summary:**

StaPO is intuitive and empirically effective at stabilizing training, but reviewers questioned whether the dual-margin objective represents a principled improvement over existing preference optimization methods or primarily acts as a heuristic regularization to prevent late-epoch degradation. It remains unclear whether its gains extend beyond improved stability and reduced sensitivity to overfitting, or whether comparable results could be achieved through careful early stopping or alternative regularization.

**Reviewer Concerns:**

Several reviewers noted that StaPO’s gains may primarily stem from improved training stability rather than a fundamentally better optimization objective, raising concerns about how much benefit it provides beyond slowing overfitting to the preference dataset. There was also concern that the dual-margin objective of StaPO lacks strong theoretical grounding.

Reviewers also raised concerns about additional hyperparameter tuning, since StaPO introduces left and right margins without a principled selection criterion, which may impact reproducibility and generalization across datasets and model families.

Empirically, reviewers felt the analysis of degenerate language behavior was limited, relying mainly on entropy and a small number of qualitative examples.

**Reviewer Scores:**

The current reviewer scores already reflect a balanced assessment of the paper’s strengths and weaknesses. While the paper is solid and not flawed, it lacks sufficient excitement or conceptual novelty, and the authors’ responses are therefore unlikely to substantially change the reviewers’ evaluations.

---

### Decision · Program_Chairs · 2026-01-26

Reject